# Modular assembly of the principal microtubule nucleator γ-TuRC

Martin Würtz [1,3], Erik Zupa [1,3], Enrico S. Atorino [1,3], Annett Neuner[1], Anna Böhler[1], Ariani S. Rahadian[1], Bram J. A. Vermeulen [1], Giulia Tonon[1], Sebastian Eustermann [2], Elmar Schiebel [1✉] & Stefan Pfeffer [1✉]

The gamma-tubulin ring complex (γ-TuRC) is the principal microtubule nucleation template in vertebrates. Recent cryo-EM reconstructions visualized the intricate quaternary structure of the γ-TuRC, containing more than thirty subunits, raising fundamental questions about γ-TuRC assembly and the role of actin as an integral part of the complex. Here, we reveal the structural mechanism underlying modular γ-TuRC assembly and identify a functional role of actin in microtubule nucleation. During γ-TuRC assembly, a GCP6-stabilized core comprising GCP2-3-4-5-4-6 is expanded by stepwise recruitment, selective stabilization and conformational locking of four pre-formed GCP2-GCP3 units. Formation of the lumenal bridge specifies incorporation of the terminal GCP2-GCP3 unit and thereby leads to closure of the γ-TuRC ring in a left-handed spiral configuration. Actin incorporation into the complex is not relevant for γ-TuRC assembly and structural integrity, but determines γ-TuRC geometry and is required for efficient microtubule nucleation and mitotic chromosome alignment in vivo.

[1] Zentrum für Molekulare Biologie der Universität Heidelberg, DKFZ-ZMBH Allianz, Im Neuenheimer Feld 282, 69120 Heidelberg, Germany. [2] European Molecular Biology Laboratory (EMBL), Heidelberg Meyerhofstraße 1, 69117 Heidelberg, Germany. [3]These authors contributed equally: Martin Würtz, Erik Zupa, Enrico S. Atorino. ✉email: e.schiebel@zmbh.uni-heidelberg.de; s.pfeffer@zmbh.uni-heidelberg.de

Microtubules are essential cytoskeletal components with a variety of functions in cellular compartmentalization, chromosome segregation, cell motility, and intracellular transport. Microtubules are formed de novo in a process termed nucleation, which is spatially and temporally highly controlled by γ-tubulin complexes, composed of γ-tubulin[1] and associated γ-tubulin-complex proteins (GCP)[2,3]. In most eukaryotes, microtubules are templated from a preassembled γ-tubulin ring complex (γ-TuRC) that determines the position and direction of microtubule growth[4,5].

For many years, our understanding of the γ-TuRC was limited to its function in microtubule nucleation and detailed mechanistic and structural insights were missing. This changed recently, when four independent studies described cryo-EM structures of the vertebrate γ-TuRC. The complex adopts the overall shape of an asymmetric left-handed spiral formed by 14 GCP-γ-tubulin heterodimers (here named spokes), each comprising one copy of γ-tubulin and one of five different paralogous GCP proteins (GCP2-6) arranged in a uniform sequence along the spiral[6–9]. All studies identified a scaffold in the lumen of the complex, termed the lumenal bridge, that unexpectedly included one molecule of actin, whose function is still elusive. The cryo-EM reconstructions allowed detailed characterization of the multivalent interactions of actin within the γ-TuRC, identifying the GCP6 N-terminus[10] as well as the γ-tubulin subunit associated with GCP3 at spoke 2 (hereafter GCP3$_{(2)}$) as the main interactors. The rest of the lumenal bridge was composed of two similar α-helical structural modules that both contained one copy of the small protein MZT1 intertwined with the N-terminus of GCP6 and GCP3, respectively. X-ray crystallography further identified the GCP5 N-terminus to be capable of forming a similar structural module in complex with MZT1[11], suggesting that the γ-TuRC has seven distinct binding sites for MZT1 (5 copies of GCP3 and one copy of GCP5 and GCP6, respectively), one on every other spoke[12].

Based on this structural molecular census of γ-TuRC components, recombinant expression systems for the human γ-TuRC were established in several independent studies[9,13,14], enabling direct genetic manipulation and thus detailed functional dissection of individual γ-TuRC components. Using recombinant systems, actin and MZT1 co-expression were observed to be essential for the structural integrity of the complex[14], and the AAA+ ATPase RUVBL1/RUVBL2 was identified as an important factor for efficient γ-TuRC reconstitution[9]. Notably, similar to the situation in yeast[15,16], human GCP2, GCP3, and two molecules of γ-tubulin were shown to be capable of forming a stable γ-tubulin small complex (γ-TuSC) unit when omitting the γ-TuRC-specific GCP paralogs (GCP4-6) from the expression system[9].

All vertebrate γ-TuRCs structurally analyzed thus far showed a virtually identical subunit architecture, indicating that precisely defined internal organizational principles drive the self-assembly of the over 30 subunits into one structurally uniform γ-TuRC. Here, we used cryo-EM to identify and characterize γ-TuRC assembly intermediates after recombinant expression of the complex, which allowed us to delineate a comprehensive stepwise assembly pathway from a stable 6-spoked core to the mature 14-spoked γ-TuRC, identifying successive stabilization and conformational locking of γ-TuSC units as central principles in the assembly process. Moreover, via targeted mutagenesis of GCP6, we could demonstrate that actin integration is not required for γ-TuRC assembly and structural integrity. However, γ-TuRC mutants unable to integrate actin showed defects in microtubule nucleation and chromosome alignment in human cells, revealing a functional role of γ-TuRC-associated actin in cell division.

## Results

**A 6-spoked core is the minimal stable intermediate in γ-TuRC assembly.** To structurally elucidate γ-TuRC assembly, we isolated human γ-TuRC from a recombinant insect cell expression system via gentle 2xFLAG-GCP5 affinity purification (see Methods; Supplementary Fig. 1), which yields microtubule nucleation-competent γ-TuRCs[13]. Subjecting the sample to cryo-EM single-particle analysis, we obtained cryo-EM reconstructions of the fully assembled γ-TuRC, but also various assembly intermediates, which enabled us to comprehensively recapitulate the assembly process (Supplementary Fig. 2, Supplementary Table 1).

To confirm the correct assembly of the γ-TuRC in our recombinant expression system, we first analyzed the consensus structure of fully assembled γ-TuRC and large assembly intermediates (10–12 spokes, Supplementary Fig. 2) to increase the sample size. For this set of particles, we obtained a cryo-EM reconstruction (Fig. 1a) at 7.5 Å global resolution (Supplementary Fig. 3a). Structural features of the cryo-EM density indicated that all GCP variants have been incorporated into the complex in the correct order and that the lumenal bridge was formed correctly, including stable integration of actin and two copies of MZT1 (Fig. 1b). Unexpectedly, we identified three previously unresolved MZT1-GCP3 modules (Fig. 1a, c; Supplementary Fig. 4a, b) and one MZT1-GCP5 module (see below) on the outer surface of the γ-TuRC, associated with the GRIP2 domains of GCP3 subunits (Fig. 1a, right). Cumulatively, these data confirm correct γ-TuRC assembly in the recombinant expression system on subunit level and—together with the previously identified MZT1-GCP3 module on spoke 14[10]—reveal binding sites for all MZT1-containing structural modules in the complex.

We next employed computational particle sorting to identify and analyze the structures of γ-TuRC assembly intermediates. Extensive 3D classification revealed a core of six γ-tubulin-GCP spokes as the minimal stable and most abundant entity in this experiment. This 6-spoke assembly intermediate was resolved to 5.3 Å resolution (Supplementary Fig. 3b), sufficient to unambiguously determine the identity and sequential arrangement of spokes based on secondary structure features of the GCP proteins. The 6-spoke assembly intermediate (Fig. 2a) comprises one γ-TuSC unit and all ring-complex specific GCP proteins in an ordered assembly of GCP2-3-4-5-4-6, corresponding to spokes 7–12 of the completely assembled complex.

**The 6-spoke intermediate is stabilized by GCP6 and the partially formed lumenal bridge.** The cryo-EM density of the 6-spoke intermediate revealed that the first module of the lumenal bridge has already been formed by recruitment of the MZT1-GCP3 module to the GCP3$_{(8)}$ GRIP1 domain (Fig. 2b). Since only one copy of GCP3 is present in this intermediate, this clearly indicates that the MZT1-GCP3 module of GCP3$_{(8)}$ (rather than any other copy of GCP3) is involved in forming the lumenal bridge. Surprisingly, the outer surface of the GCP3$_{(8)}$ GRIP2 domain is associated with an additional MZT1-containing module (Fig. 2c). As the only MZT1-GCP3 module present in the 6-spoke intermediate was already stably recruited into the lumenal bridge, MZT1-GCP5 and MZT1-GCP6 were the only other candidates to represent this additional MZT1 module. Docking X-ray structures of human GCP5- and GCP6-containing MZT1 modules[11] (Supplementary Fig. 4c) allowed us to identify the unassigned density segment associated with the GCP3$_{(8)}$ GRIP2 domain as the MZT1-GCP5 module (Fig. 2c). Notably, GCP3$_{(8)}$ and the adjacent GCP4$_{(9)}$ subunit create a structurally unique binding site for this MZT1-GCP5 module (Fig. 2c), which likely explains the specific binding of MZT1-GCP5 to the outer surface of GCP3$_{(8)}$, while MZT1-GCP3$_{(8)}$ remains available to form the first module of the lumenal bridge (Fig. 2d). Having located the MZT1-GCP5 module on GCP3$_{(8)}$ and the MZT1-GCP6 module as part of the lumenal bridge, the four additional MZT1-containing modules resolved on

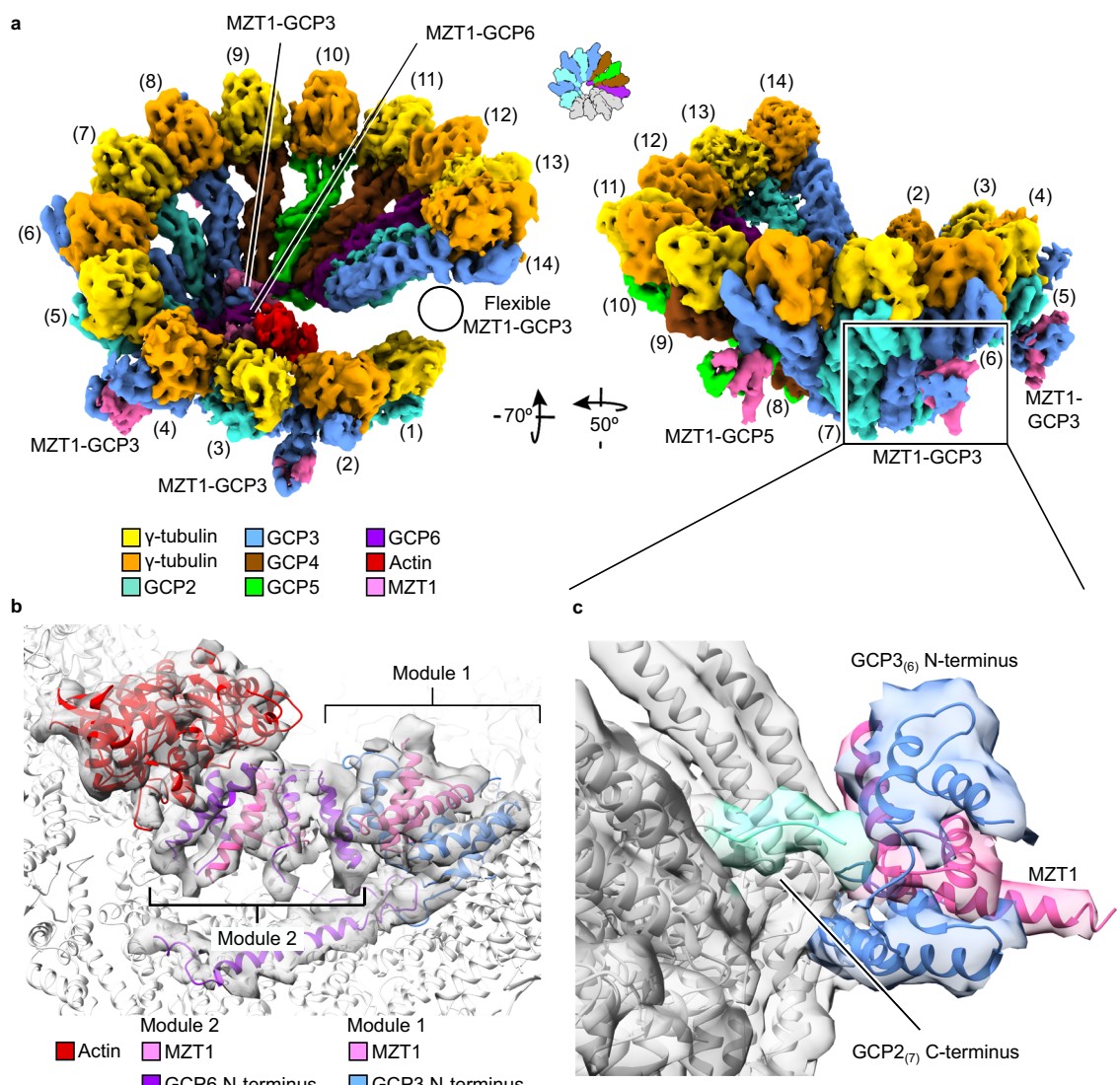

**Fig. 1 Missing MZT1 modules bind on the outer surface of the recombinant human γ-TuRC. a** Two views on the cryo-EM reconstruction of the recombinant human γ-TuRC. Spokes are numbered and coloring is indicated. The positioning of all MZT1 modules is highlighted. The position of the MZT1 module on GCP3$_{(14)}$ was not visible in this structure due to substoichiometry and structural heterogeneity, but is resolved in other reconstructions at the indicated position (black circle). Substoichiometry of the γ-TuSC$_{(1,2)}$ and γ-TuSC$_{(13,14)}$ units in this reconstruction is indicated by grey coloring in the schematic representation. **b** Structure and composition of the lumenal bridge. Atomic model (PDB 6X0U) superposed to the segmented density of the lumenal bridge. Coloring as indicated. **c** Cryo-EM density of the MZT1-GCP3 module associated with GCP3$_{(6)}$. An atomic model of the MZT1-GCP3 module from the lumenal bridge (PDB 6X0U) was docked as a rigid body. MZT1-GCP3 modules directly interact with the adjacent γ-TuSC units via the GCP2 C-terminus.

GCP3$_{(2)}$, GCP3$_{(4)}$, GCP3$_{(6)}$, and GCP3$_{(14)}$ of larger assembly intermediates and the fully assembled γ-TuRC represent the remaining four MZT1-GCP3 modules.

By extending the atomic models for GCP5 and GCP6 in a previously published high-resolution cryo-EM density of the γ-TuRC[10] (Supplementary Fig. 5a), we identified the N-terminal extension of the GCP6 GRIP1 domain (282–351; GRIP1-NTE) as a major determinant in stabilizing the MZT1-GCP3$_{(8)}$ module in the lumen of the complex, by binding to the N-terminal helix of MZT1 (Fig. 2e; Supplementary Fig. 5b, c), and thus indirectly in recruiting the γ-TuSC$_{(7,8)}$ unit that is part of the 6-spoke assembly intermediate. The overall stabilizing role of the GCP6 GRIP1-NTE is further supported by direct contacts of GCP6 with the GCP5$_{(10)}$ GRIP1-NTE and the GRIP1 domains of GCP4$_{(9)}$, GCP5$_{(10)}$, and GCP4$_{(11)}$, effectively bridging all GCP subunits of the intermediate (Fig. 2e; Supplementary Fig. 5b–e). Taken

together, we conclude that the GCP6 GRIP1-NTE plays a central role in the formation of the stable 6-spoke intermediate, providing a structure-based rationale for the effects of GCP6 truncations on γ-TuRC assembly previously observed in sucrose gradient sedimentation experiments[17].

Previous studies using salt fragmentation analysis of native γ-TuRC indicated that GCP4, GCP5, and GCP6 can form a stable complex[6]. To test this possibility, we specifically omitted GCP2 and GCP3 from the expression system, reconstituting γ-TuRC$^{ΔGCP2/3}$ (Supplementary Fig. 6a). Coomassie-stained SDS-PAGE and immunoblot analysis indicated successful copurification of GCP4, GCP6 and γ-tubulin with 2xFLAG-GCP5 (Supplementary Fig. 6b, c) and negative stain EM in conjunction with 2D class averaging showed reconstitution of a 4-spoke entity (Supplementary Fig. 6d, e). Using cryo-EM analysis, we obtained a reconstruction of this 4-spoke assembly at 7.8 Å global

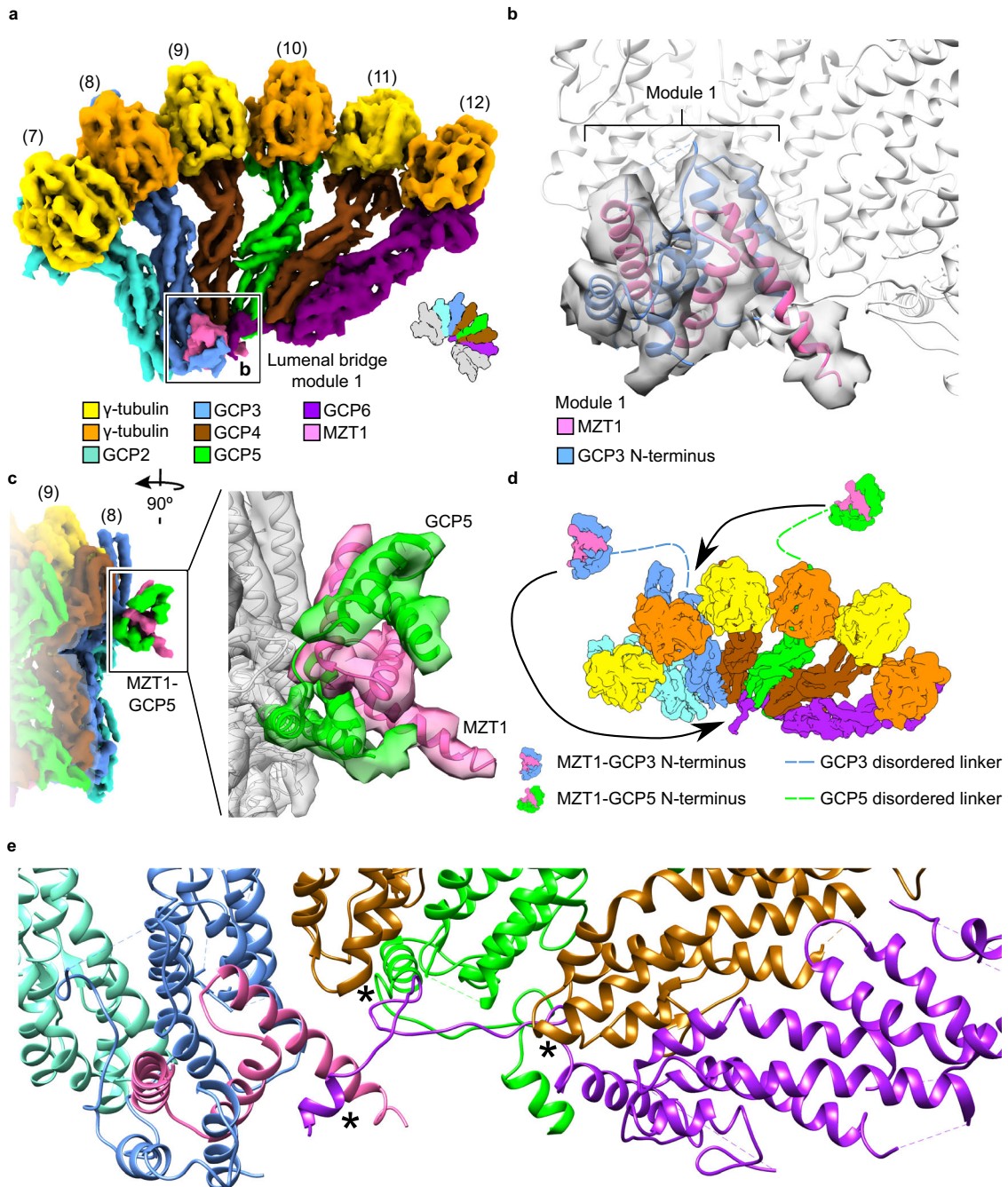

**Fig. 2 γ-TuRC assembly starts from a GCP6-stabilized 6-spoke intermediate. a** Molecular architecture of the 6-spoke assembly. Numbering of spokes according to the full γ-TuRC and coloring of the components as indicated. Substoichiometry of the $\gamma\text{-TuSC}_{(5,6)}$ and $\gamma\text{-TuSC}_{(13,14)}$ units in this reconstruction is indicated by grey coloring in the schematic representation. **b** Atomic model (PDB 6X0U) superposed to the segmented density of lumenal bridge module 1. The lumenal bridge components are colored as indicated. Zoom as indicated in panel (**a**). **c** Zoom on the MZT1-GCP5 module. Atomic model of the MZT1-GCP5 module (PDB 6L81) superposed to the α-helical bundle on the outer surface of the 6-spoke assembly. **d** The MZT1-GCP5 module binds to the outer surface of $\text{GCP3}_{(8)}$, while $\text{MZT1-GCP3}_{(8)}$ remains available to form the first module of the lumenal bridge. **e** Atomic model of the γ-TuRC including newly built segments of the GCP5 and GCP6 N-terminal regions. Zoom focusing on the interaction of the GCP6-NTE with both copies of GCP4, GCP5, and the lumenal bridge components. Interactions are indicated with asterisks. Coloring as in (**a**).

resolution (Supplementary Figs. 3c and 7a–d), which was sufficient to unambiguously identify the GCP variant order as GCP4-5-4-6, based on the length of resolved GCP N-termini and specific GCP extensions (Supplementary Fig. 7e, f). The GCP variant order in the 4-spoke intermediate thus recapitulates the GCP sequence of spokes 9–12 in the fully assembled γ-TuRC, indicating that transient formation of a GCP4-5-4-6 intermediate,

followed by rapid and stable integration of one γ-TuSC unit, likely precedes the formation of the 6-spoke intermediate during γ-TuRC assembly.

**Successive recruitment and conformational locking of preformed γ-TuSC units expands the 6-spoked core on the $\text{GCP2}_{(7)}$-facing side.** Having characterized the start and

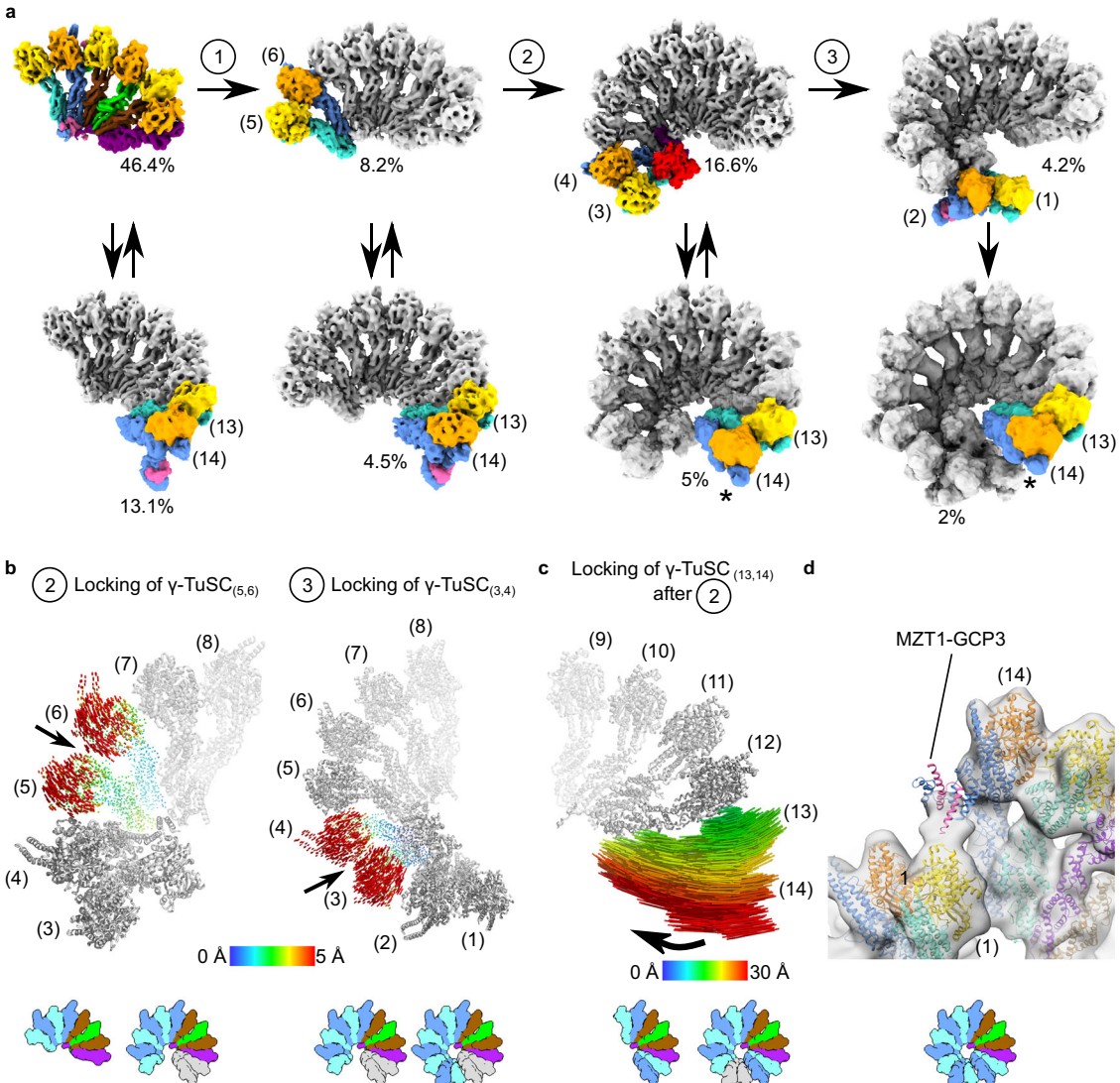

**Fig. 3 The 6-spoked core is expanded by successive recruitment and conformational locking of γ-TuSC units. a** Cryo-EM reconstructions of γ-TuRC assembly intermediates arranged into a possible pathway for γ-TuRC assembly. Newly added γ-TuSC units are colored as in Fig. 1a, while the rest of the complex is depicted in grey. Newly added spokes are numbered according to the fully assembled γ-TuRC. Reconstructions in which the γ-TuSC(13,14) unit is in the locked conformation are indicated by an asterisk. Alternative assembly pathways and intersections between different pathways are possible. Percentage is given for homogeneous sets of particles representing distinct assembly states. **b** Conformational locking of γ-TuSC units, as represented by vectors linking the Cα atoms in both conformations. Vectors are colored according to the scale bar. Directionality of the motion is represented by an arrow. Icons indicate the set of reconstructions used for the comparison. **c** Slide-in motion of the γ-TuSC(13,14) unit after the formation of the lumenal bridge. Representation as in panel (**b**). **d** Position of MZT1-module at the locked γ-TuSC(13,14). MZT1-module binds to the γ-tubulin associated with GCP2(1).

endpoints of γ-TuRC assembly, we next turned toward the structural analysis of assembly intermediates that link these two stages (Supplementary Fig. 2). These intermediates arise from the modular and successive expansion of the 6-spoked GCP2-3-4-5-4-6 core with preformed γ-TuSC units (Fig. 3a). Notably, expansion by individual heterodimeric γ-tubulin-GCP spokes was never observed, indicating that exclusively preformed γ-TuSC units are being added.

On the GCP2(7)-facing side of the 6-spoke assembly, the expansion phase starts with the recruitment of one γ-TuSC unit, forming spokes 5 and 6 of the fully assembled complex. Notably, the conformation of GCPs in this newly recruited γ-TuSC(5,6) unit differs when compared to the fully assembled complex. In particular, the GRIP2 domains and the associated γ-tubulin copies are displaced from their canonical position (Fig. 3b). In the next step of complex assembly, a second γ-TuSC unit is recruited

to the GCP2(5)-facing side of the assembly intermediate, occupying positions 3 and 4 of the fully assembled γ-TuRC. Interestingly, during this step of complex assembly, the GRIP2 domains of the γ-TuSC(5,6) unit are repositioned by ~5 Å towards the helical axis of the γ-TuRC and become locked in their final position (Fig. 3b). It is only during this stage of the expansion phase when the second module of the lumenal bridge, comprising the MZT1-GCP6 module and actin (Fig. 1b), is stably integrated into the assembly intermediate (Fig. 3a), indicating a stepwise and interdependent assembly process of γ-TuRC components similar to multi-protein complex assembly in eukaryotic chromatin remodelers[18] or ribosome biogenesis[19].

Upon recruitment of the terminal γ-TuSC unit to the GCP2(3)-facing side of the assembly intermediate, the conformation of the γ-TuSC(3,4) unit is locked in a manner similar to γ-TuSC(5,6) (Fig. 3b). Different from γ-TuSC(3,4) and γ-TuSC(5,6), the terminal

$\gamma$-TuSC$_{(1,2)}$ unit is already integrated upon its initial binding in a conformation that matches the structural organization in the fully assembled complex. Collectively, these data suggest that the recruitment and subsequent 'conformational locking' of individual $\gamma$-TuSC units assumes a central role during $\gamma$-TuRC assembly.

**A lumenal bridge-dependent two-step mechanism integrates $\gamma$-TuSC$_{(13,14)}$ and completes complex assembly.** Each of the assembly intermediates described above exists in two structural variants, including or lacking the $\gamma$-TuSC$_{(13,14)}$ unit (Fig. 3a). The almost constant ratio between $\gamma$-TuSC$_{(13,14)}$-containing and -lacking intermediates for each assembly step argues for repeated association and dissociation of $\gamma$-TuSC$_{(13,14)}$ in an equilibrium reaction.

Unexpectedly, the $\gamma$-TuSC$_{(13,14)}$ unit undergoes a striking repositioning of up to 30 Å relative to the directly adjacent GCP6 subunit at a specific step of complex assembly (Fig. 3c), when recruitment of $\gamma$-TuSC$_{(3,4)}$ completes the formation of the lumenal bridge. Comparison of two distinct conformational states, right before and after the formation of the lumenal bridge, indicates that the $\gamma$-TuSC$_{(13,14)}$ unit slides along the GCP6$_{(12)}$-GCP2$_{(13)}$ GRIP1 domain interface towards the lumenal bridge, resulting in an altered, more ring-like overall geometry of the $\gamma$-TuRC. In this new, 'locked' position, the $\gamma$-TuSC$_{(13,14)}$ unit can directly bind to the $\gamma$-TuSC$_{(1,2)}$ unit by virtue of the MZT1-GCP3$_{(14)}$ module (Fig. 3d), which stabilizes the $\gamma$-TuSC$_{(13,14)}$ unit on the complex and thus may prevent it from dissociating again. This indicates a two-step mechanism for completion of complex assembly and ring closure, involving first, positional locking and, second, stabilization of the $\gamma$-TuSC$_{(13,14)}$ unit by $\gamma$-TuSC$_{(1,2)}$. These two steps may reflect first reversible and subsequently more stable binding of the $\gamma$-TuSC$_{(13,14)}$ unit.

Collectively, the observed intermediate steps can be arranged into a pathway for modular assembly of the $\gamma$-TuRC, in which the 6-spoke intermediate is expanded by successive recruitment of $\gamma$-TuSC units on the GCP2$_{(7)}$-facing side while the $\gamma$-TuSC$_{(13,14)}$ unit repeatedly associates and dissociates (Fig. 3a) until it is stabilized in the final step of complex assembly.

**Selective stabilization of $\gamma$-TuSC units underlies uniform $\gamma$-TuRC architecture.** Recombinant $\gamma$-TuSC units have been observed to self-interact and spontaneously form ring-like oligomers of varying sizes under mild cross-linking conditions[9]. In an attempt to understand the mechanism underlying the defined expansion of the 6-spoke intermediate by exactly four $\gamma$-TuSC units into a uniform 14-spoked $\gamma$-TuRC (Fig. 3a), we analyzed the oligomerization behavior of $\gamma$-TuSC units in detail. For this purpose, we purified $\gamma$-TuSC units from a recombinant insect cell expression system via GCP3-2xFLAG affinity purification and anion exchange chromatography (see Methods; Supplementary Fig. 8).

We analyzed the oligomeric state of $\gamma$-TuSC units by negative stain EM and 2D class averaging in a dilution series of the purified complex (Supplementary Fig. 9a–d). Both the relative fraction of oligomerized $\gamma$-TuSC units and the oligomer size increased in a concentration-dependent manner (Supplementary Fig. 9c, d), confirming that $\gamma$-TuSC units indeed have a latent capability to self-interact and spontaneously form oligomers. However, while these data indicate that a high concentration of $\gamma$-TuSC units is required for spontaneous oligomerization (>0.5 μM), a plethora of biochemical data[6–9] have established that the $\gamma$-TuRC from various sources remains intact after isolation and purification, entirely independently of a pool of 'free' $\gamma$-TuSC units to drive the equilibrium towards oligomer

assembly. This clearly indicates that additional binding interfaces provided by components other than the $\gamma$-TuSC are important in selectively stabilizing $\gamma$-TuSC units after recruitment to the growing $\gamma$-TuRC subcomplex. The GCP6 GRIP1-NTE is a good candidate to partially fulfil such a stabilizing function, as it binds to all $\gamma$-TuSC units on the GCP4$_{(9)}$-facing side of the complex (Supplementary Fig. 10).

**Actin is not required for $\gamma$-TuRC assembly and structural integrity.** Based on previously available structural data of the fully assembled $\gamma$-TuRC, $\gamma$-TuRC-associated actin had been suggested to play a major structural role in the complex[6–8,10,14], in particular in recruiting the $\gamma$-TuSC$_{(1,2)}$ unit that directly interacts with actin (Fig. 1a, b). To test this hypothesis, we aimed to specifically inhibit the incorporation of actin into the $\gamma$-TuRC by perturbing its binding site. For this purpose, we analyzed the interface between actin and the remaining $\gamma$-TuRC components in detail and confirmed the two N-terminal helices of GCP6 that are part of the lumenal bridge as the predicted main interactors[10] (Supplementary Fig. 11a). To biochemically test whether the GCP6 N-terminus is sufficient to recruit actin into the $\gamma$-TuRC, we co-expressed the first 126 amino acid residues of *TUBGCP6* (GCP6$_{N126}$-His$_8$) together with *FLAG-MZT1* in *E. coli* to recombinantly reconstitute the MZT1-GCP6 module of the lumenal bridge. Co-IP experiments against the C-terminal His-Tag on the GCP6 construct showed that the MZT1-GCP6 module was able to pull down actin (Fig. 4a), while the GCP6$_{N126}$-His$_8$ without FLAG-MZT1 failed to do so (Fig. 4a). This indicates that the correctly folded MZT1-GCP6 module forms a strong interaction with actin, likely sufficient to recruit it to the $\gamma$-TuRC, independent of other interactions in the complex. Next, we truncated the two actin-interacting GCP6 $\alpha$-helices from the construct (GCP6$_{N57-126}$) and co-expressed it with *FLAG-MZT1*. We no longer observed Co-IP of the MZT1-GCP6$_{N57-126}$ module with actin (Fig. 4a), indicating that the first two GCP6 $\alpha$-helices are the main contributors to actin binding, as initially suggested by analysis of previously available structures[10].

To study the impact of inhibited actin integration on the structure of $\gamma$-TuRC, we replaced full-length GCP6 in the recombinant expression system by an N-terminally truncated form of GCP6 ($\Delta$N56-GCP6), reconstituted $\gamma$-TuRC$^{\Delta N56\text{-}GCP6}$ (Supplementary Fig. 12) and subjected the complex to cryo-EM analysis. During computational sorting of cryo-EM particles, we identified similar $\gamma$-TuRC assembly intermediates as for wild-type $\gamma$-TuRC (Supplementary Fig. 11), but unexpectedly also fully assembled 14-spoke $\gamma$-TuRC$^{\Delta N56\text{-}GCP6}$ complexes. We obtained a cryo-EM reconstruction of fully assembled $\gamma$-TuRC$^{\Delta N56\text{-}GCP6}$ (Fig. 4b) at 7.1 Å global resolution (Supplementary Figs. 3o and 10b), which was highly similar to the wild-type complex. However, as expected, actin was not incorporated into the $\gamma$-TuRC$^{\Delta N56\text{-}GCP6}$, as indicated by the absence of cryo-EM density for this component (Fig. 4c; Supplementary Fig. 11b, c). Surprisingly, the remaining segments of the lumenal bridge were still well-structured, allowing for detailed dissection of the MZT1-containing modules in the lumenal bridge. While the MZT1-GCP3$_{(8)}$ module forming the basis of the lumenal bridge was structurally identical to the wild-type complex, no cryo-EM density could be observed for the first two N-terminal GCP6 $\alpha$-helices, as expected (Fig. 4c). Notably, all three $\alpha$-helices of the GCP6-associated MZT1 copy were well-resolved, indicating that the first two N-terminal GCP6 $\alpha$-helices are dispensable for stable folding of the MZT1-GCP6 module.

In conclusion, our Co-IP experiments in conjunction with cryo-EM analysis of $\gamma$-TuRC$^{\Delta N56\text{-}GCP6}$ indicate that the GCP6 N-terminus plays a major role in actin-binding and unexpectedly

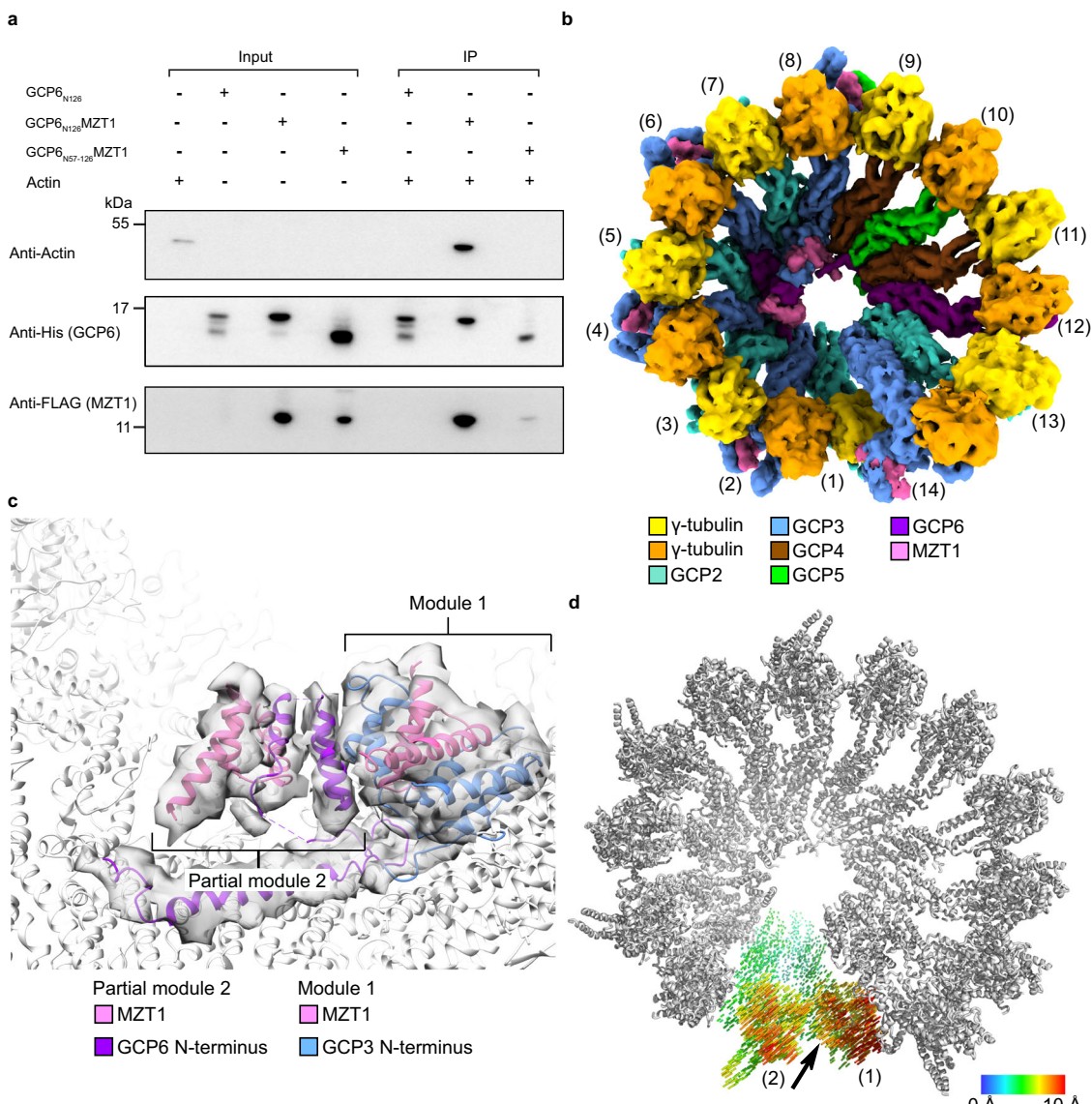

**Fig. 4 Actin is not required for γ-TuRC assembly but impacts on γ-TuRC geometry. a** The very N-terminus of GCP6 is crucial for actin binding. Immunoblot analysis of an actin IP experiment against His tagged, N-terminal fragments of GCP6 (GCP6$_{N126}$-His$_8$; GCP6$_{N57-126}$-His$_8$), co-expressed with FLAG-MZT1 in *E. coli*. Input and IP samples were probed against anti-actin, anti-His (GCP6) and anti-FLAG (MZT1) antibodies. Sections of representative immunoblots of one experiment are shown. IP experiments were performed in (*n* = 3) independent experiments and source data are provided as a Source Data file. **b** Cryo-EM reconstruction of γ-TuRC$^{ΔN56-GCP6}$ deficient in actin binding. Coloring as indicated. Spokes are numbered. **c** Structure and composition of the lumenal bridge in γ-TuRC$^{ΔN56-GCP6}$. Atomic model of resolved components (PDB 6X0U) superposed to the segmented density of the lumenal bridge. The structure is devoid of cryo-EM density for the two N-terminal helices of GCP6 and actin. Coloring as indicated. **d** Effect of actin integration on γ-TuRC geometry, as represented by vectors linking the Cα atoms in both conformations. Vectors are colored according to scale bar. Directionality of motion is represented by an arrow.

show that actin is not required for assembly or structural integrity of the γ-TuRC.

**Inhibiting actin incorporation impacts on γ-TuRC geometry and negatively affects microtubule nucleation activity in vivo.** Having excluded a role for actin in γ-TuRC assembly and structural integrity, we next aimed to explore if actin participates in γ-TuRC-mediated microtubule formation. γ-tubulin complexes have been proposed to act as structural templates during de novo formation of microtubules, suggesting that the spatial arrangement of GCP-associated γ-tubulin molecules plays a decisive role in the efficiency of the process[20,21]. We therefore compared the overall geometry of wild-type and actin binding-deficient γ-TuRC and observed that the arrangement

and conformation of spokes 3–14 is very similar in both complexes. In contrast, the positioning of the γ-TuSC$_{(1,2)}$ GRIP2 domains is altered in the actin binding-deficient γ-TuRC and the two respective γ-tubulin subunits are displaced away from the helical axis (Fig. 4d), resulting in a γ-tubulin positioning that is less compatible with the almost ideal helical arrangement of αβ-tubulin units in the microtubule lattice. The structural basis for repositioning of the γ-TuSC$_{(1,2)}$ GRIP2 domains is likely to be the interaction between the actin D-loop and γ-tubulin at spoke 2[6].

Having observed changes in γ-TuRC geometry dependent on the integration of actin, we next tested the nucleation efficiency of γ-TuRC$^{ΔN56-GCP6}$ using two complementary in vitro microtubule nucleation assays. In the first assay, nucleation-dependent batch

fluorescence was measured over a range of γ-TuRC concentrations (Supplementary Fig. 12g). In the second assay, nucleated microtubules were pelleted and counted on light microscopy images to achieve a more direct readout at early time points after starting the nucleation reaction (Supplementary Fig. 12h). In both assays, γ-TuRC^ΔN56-GCP6 nucleated microtubules to a similar extent as the recombinant wild-type γ-TuRC and thus, under the nucleation conditions of the in vitro experiments, an impact of γ-TuRC-associated actin on microtubule formation could not be observed. This is in agreement with single-molecule TIRF experiments[14], in which a similar microtubule nucleation efficiency was observed for wild-type γ-TuRC and structurally highly impaired MZT1- and actin-deficient γ-tubulin complexes, recapitulating only half of the γ-tubulin ring. This outcome indicates that the microtubule nucleation process in vitro is rather indiscriminative with respect to the structure of the γ-tubulin-complex template, but changes in nucleation activity in vivo cannot be excluded due to the involvement of additional cofactors, such as XMAP215[7,22,23].

Therefore, we assessed what effect the actin binding-deficient γ-TuRC has on microtubule nucleation and cell viability under physiological conditions in vivo. In this analysis we included the N-terminal deletion construct structurally characterized using cryo-EM (ΔN56-GCP6; Fig. 4; Supplementary Fig. 12), but also several constructs with combinations of point mutations within the GCP6 N-terminal region predicted to affect the interaction with actin (see Methods, Supplementary Fig. 11a), aiming to prevent actin integration with minimal structural perturbation of γ-TuRC components. We tested the ability of these point mutation GCP6 constructs to pull down actin in vitro, as described above, and confirmed that all mutant GCP6 were deficient in actin binding, while they still interacted with MZT1 (Supplementary Fig. 12i).

Next, we stably integrated *TUBGCP6-C-terminal-FLAG* constructs for regulated expression in RPE1-hTERT cells. To assess the microtubule nucleation behavior of the γ-TuRC with mutant GCP6 versions in vivo, we conducted a microtubule regrowth assay in RPE1-hTERT cells (Fig. 5a; Supplementary Fig. 13a). After cold-induced microtubule depolymerization, we quantified the number and length of renucleated microtubules at the centrosome, where the γ-TuRC is expected to be the principal microtubule nucleator. Depletion of endogenous GCP6 by siRNA-mediated gene silencing strongly decreased microtubule regrowth (Fig. 5a–c; Supplementary Fig. 13a–c), but transient expression of siRNA-resistant wild-type *TUBGCP6* could rescue microtubule regrowth to the levels observed in untreated cells or cells treated with noncoding siControl. In contrast, microtubule renucleation was strongly affected after transient expression of siRNA-resistant *TUBGCP6* mutants (Fig. 5a–c; Supplementary Fig. 13a–c). Cells expressing mutant GCP6, especially the variants containing all point mutations (ALL mut) or lacking the N-terminal helices (ΔN56-GCP6), nucleated less and shorter microtubules than in the siControl or the rescue condition (Fig. 5a–c), indicating slower kinetics of microtubule nucleation mediated by actin binding-deficient γ-TuRC in vivo. To exclude an influence of abnormal *TUBGCP6* mutant expression levels on microtubule nucleation, we established that mutant GCP6-FLAG proteins were as abundant at centrosomes as wild-type GCP6-FLAG proteins (Supplementary Fig. 13c). Moreover, mutant γ-TuRC^ΔN56-GCP6 purified from HEK T293 cells via ΔN56-GCP6-FLAG pulldown (see Methods) was analysed via negative stain EM and 2D class averaging (Supplementary Fig. 13e), demonstrating that γ-TuRC^ΔN56-GCP6 is intact in the human cell line model and its overall structure is indistinguishable from γ-TuRC^ΔN56-GCP6 purified from the

recombinant insect cell expression system (Supplementary Fig. 12e, f). Thus, actin binding-deficient GCP6 mutant γ-TuRCs are correctly recruited to the centrosome and structurally intact in human cells, but show reduced microtubule nucleation activity in vivo.

**Mitotic spindle assembly and chromosome alignment are slowed down after inhibiting actin incorporation into the γ-TuRC.** 6Microtubule dynamics are crucial for mitotic spindle assembly, with direct effects on overall mitotic duration, chromosome alignment efficiency, and spindle shape[24]. To analyze how these parameters were affected by inhibiting actin incorporation into the γ-TuRC under more controlled conditions, we generated endogenous *ΔN-TUBGCP6* (deletion of the first 60 AA, referred to as *ΔN-GCP6*) cell lines and two independently constructed clones were used for analysis (Supplementary Fig. 13d, f, g). We observed a significant defect in microtubule nucleation activity and regrowth kinetics (Supplementary Fig. 13k, l), confirming the results shown in the siRNA experiment (Fig. 5a–c; Supplementary Fig. 13a). In *ΔN-GCP6* cells, we found no significant reduction in the γ-tubulin and GCP6 levels at the centrosome (Supplementary Fig. 13h–j), excluding that the marked differences in microtubule nucleation were caused by reduced γ-TuRC recruitment. In order to follow the mitotic progression, we generated cell lines from both *TUBGCP6* wild-type and *ΔN-GCP6* clones, in which *mNeonGreen-LMNB1* (encoding mNeonGreen-LaminB1) and *TUBG1-mRuby2* (encoding γ-tubulin-mRuby2) were constitutively expressed. These two markers were exploited to measure mitosis length (time) from centrosome separation in prophase to nuclear envelope reformation in telophase-cytokinesis transition. Live-cell imaging of these cell lines showed a 30–40 min delay of mitotic timing in *ΔN-GCP6* clones in comparison to wild-type cells (Fig. 5d, e; Supplementary Movies 1–3). Thus, the microtubule nucleation defect of γ-TuRCs in *ΔN-GCP6* cells hinders mitotic progression.

Mitosis duration mostly depends on the efficiency of spindle assembly and chromosome alignment[25]. The prolonged mitosis defect in the *ΔN-GCP6* cells therefore led us to hypothesize that the defects in the microtubule nucleation kinetics we observed in interphase might also impact on spindle formation and/or correct positioning of the chromosomes into the metaphase plate. Upon imaging of cells in which the chromosomes were aligned or were close to alignment in the metaphase plate (from late pro-metaphase to metaphase), we observed an increased, persistent accumulation of the spindle assembly checkpoint protein BubR1, a marker for defective kinetochore-microtubule attachments[26], on individual kinetochores in the *ΔN-GCP6* clones in comparison to wild-type cells (Fig. 5f, g). This indicates a delay of chromosome positioning due to defective kinetochore-microtubule attachments and thus explains the cause of the mitotic delay (Fig. 5d, e). Moreover, during these phases of mitosis, we observed detachment of the centrosomes from the spindle poles in the *ΔN-GCP6* cells (Fig. 5f, h). Consistently, these phenotypes were also prominent in siRNA depletion experiments (Supplementary Fig. 13m, n). As a consequence of these defects, we observed an increased frequency of chromosome misalignment (Fig. 5i, j; Supplementary Fig. 13o), which could lead to defective genome segregation into the new daughter cells.

In conclusion, we could show that although actin is not required for γ-TuRC assembly and structural integrity, loss of actin from the γ-TuRC hampers microtubule nucleation in vivo and slows down the mitotic progression by affecting both spindle microtubule organization and chromosome segregation/positioning.

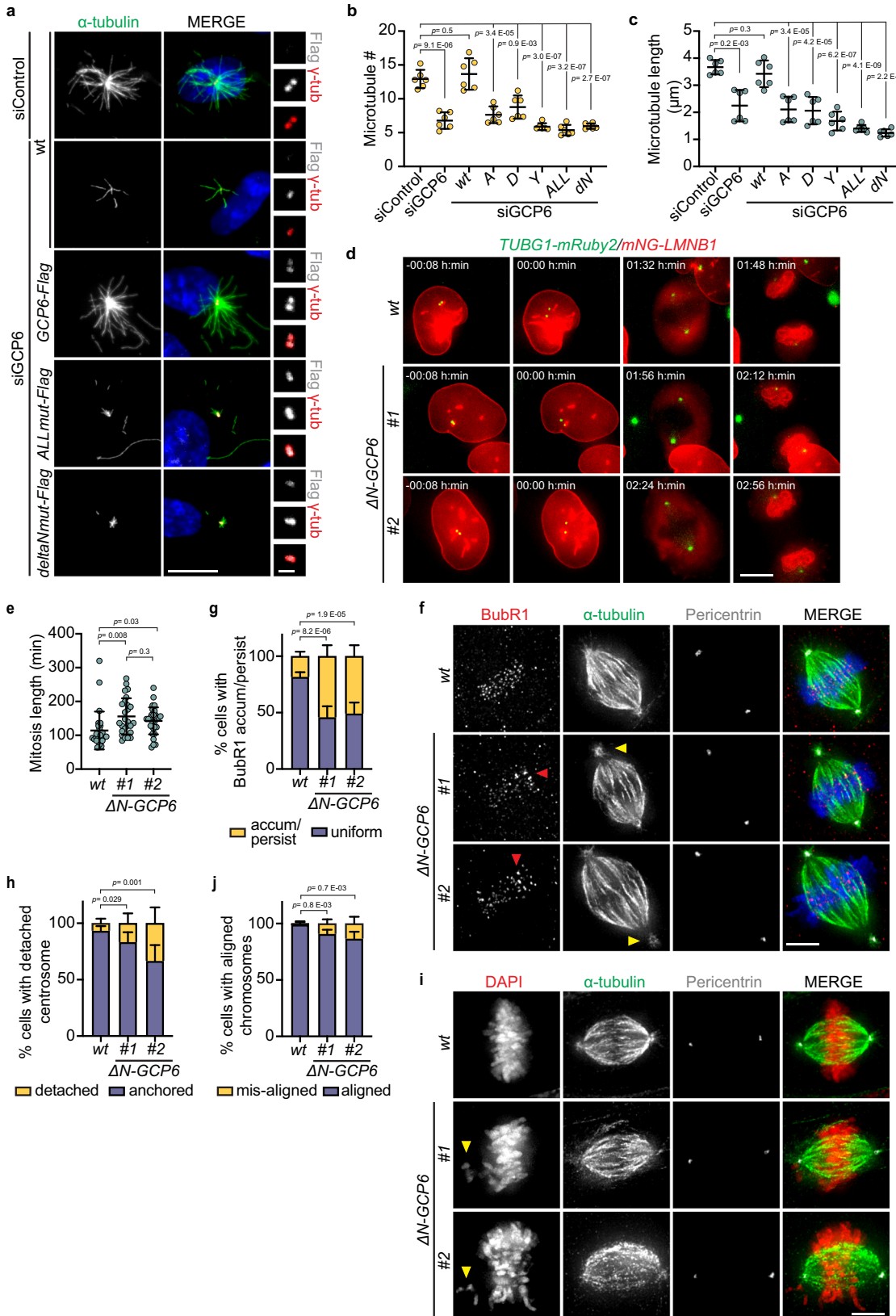

## Discussion

It is a fundamental question in biology how large multi-protein complexes are assembled from dozens of subunits to form functional and structurally uniform units—a notion that re-emerged when recent cryo-EM structures of the vertebrate γ-TuRC revealed its defined and highly ordered composition of paralogous proteins in an intricate asymmetric arrangement. In this study, we provide a mechanistic understanding for γ-TuRC assembly and the principles underlying its defined architecture. We delineate a modular assembly pathway (Fig. 6) based on the structural molecular census of assembly intermediates present in our cryo-EM data. Importantly, all of these assembly

**Fig. 5 In vivo loss of γ-TuRC-associated actin slows microtubule nucleation kinetics and prolongs mitotic progression. a–c** In vivo microtubule regrowth assay in RPE1 cells. **a** IF of microtubule asters nucleated from the centrosome (γ-tubulin) in cells treated with siRNA and in which *TUBGCP6-C-terminal-FLAG* (*GCP6-Flag*) wild-type (WT) and mutant constructs (*A* (R35A, K38A, K39A); *D* (R35D, K38D, K39E); *Y* (Y42A, F46A); *ALL* (R35D, K38D, K39E, Y42A, F46A); *dN* (ΔN56-GCP6)) were expressed (Flag). Coloring as indicated, representative images of *Amut-Flag*, *Dmut-Flag*, and *Ymut-Flag* are shown in Supplementary Fig. 13a. Average number (**b**) and average size (**c**) of microtubules nucleated from the centrosome in the samples in (**a**). **d** Representative images of live-cell imaging of RPE1 *WT* and *ΔN-GCP6* mitotic cells (Supplementary Movies 1–3) at frames showing cohered centrosomes, centrosome separation (time point 00:00 h:min), metaphase spindle and nuclear envelope reformation, coloring as indicated. **e** Quantification of intervals from centrosome separation to nuclear envelope reformation (**d**). **f** IF of metaphase *WT* and *ΔN-GCP6* mitotic cells in which BubR1 was labeled to detect its accumulation or persistence (red arrowheads) on chromosomes stained with DAPI, coloring as indicated. Centrosomes at the spindle poles were labeled with the centrosome marker pericentrin. Yellow arrowheads show detached centrosomes from the spindle pole. **g** Quantification of metaphase cells (%) in which BubR1 accumulated/persisted on centromeres non-uniformly (**f**), coloring as indicated. **h** Quantification of metaphase cells (%) in which centrosomes detached from the spindle pole (**f**), coloring as indicated. **i** IF of metaphase *WT* and *ΔN-GCP6* mitotic cells in which the chromosomes were aligned or close to the alignment in the metaphase plate. Yellow arrowheads point to mis-aligned chromosomes. **j** Quantification of metaphase cells (%) in which chromosomes did not align in the metaphase plate (**i**), coloring as indicated. (**a** and **d**, scale bars: 10 μm; **f** and **i**, scale bars 5 μm; magnification scale bars in **a**: 1 μm; **b**, **c**, **e**, **g**, **h** and **j** data are presented as mean ± s.d.; all statistics were derived from two-tail unpaired t-test analysis of: 3 replicates of 2 independent experiments (*n* = 6, **b**, **c**, **g**, **h** and **j**) and **e** (*WT n* = 27 cells, *ΔN-GCP6 #1 n* = 26 cells, and *ΔN-GCP6 #2 n* = 30 cells; all from 4 independent data acquisitions). Source data are provided as a Source Data file.

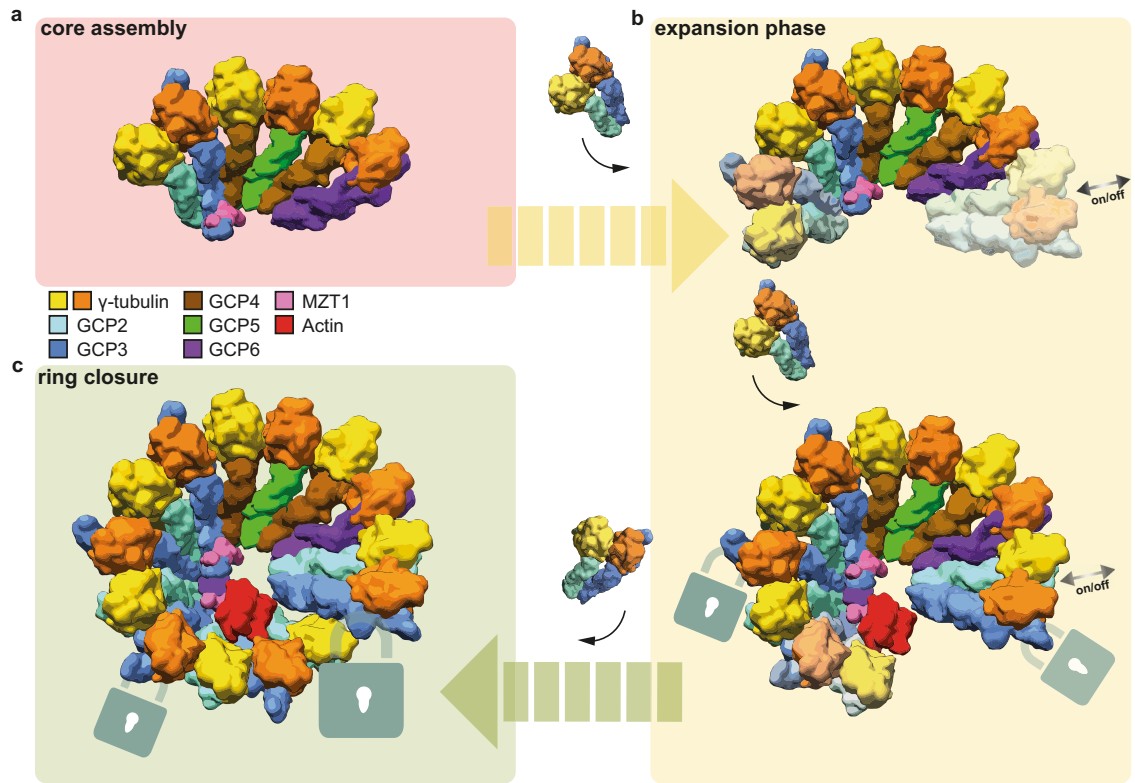

**Fig. 6 Mechanistic model for modular assembly of the human γ-TuRC. a** The initial step of γ-TuRC assembly is the formation of the 6-spoked stable core consisting of GCP2-3-4-5-4-6, in which the MZT1-GCP3 module is stabilized via the GCP6 GRIP1-NTE. Coloring as indicated. **b** Assembly continues (yellow arrow) via successive addition of γ-TuSC units. In this expansion phase, γ-TuSC units are conformationally locked (schematic lock symbol) when neighboring γ-TuSC units associate to the assembly intermediates (lighter color). During this process, the γ-TuSC$_{(13,14)}$ unit can repeatedly associate and dissociate. It is repositioned as soon as the lumenal bridge has formed. **c** Expansion ends (green arrow) in-ring closure, where the γ-TuSC$_{(1,2)}$ unit is integrated, stabilizes terminally the γ-TuSC$_{(13,14)}$ unit and locks the fully assembled 14-spoked complex.

intermediates represent subcomplexes of the γ-TuRC directly reflecting the molecular architecture of the fully assembled complex. This clearly indicates that the assembly intermediates characterized in this study represent bona fide steps in cellular biogenesis of the γ-TuRC, as frequently observed using biochemical approaches, for instance sucrose gradient analysis[2,17,27] or salt fragmentation experiments[6,17].

The assembly process characterized by cryo-EM analysis comprises of an initial core assembly step (Fig. 6a), a modular expansion phase (Fig. 6b) and a final 'ring closure' step that completes complex

assembly (Fig. 6c). Assembly starts with the formation of a 4-spoke GCP4-5-4-6 subcomplex that rapidly and quantitatively integrates one γ-TuSC unit, indicating strong cooperativity for GCP2-3-4-5-4-6 formation[28]. The resulting 6-spoke GCP2-3-4-5-4-6 intermediate can thus be considered the first stable assembly intermediate in cells, underlying all downstream steps of γ-TuRC assembly (Fig. 6a). The initial 4-spoke and 6-spoke assembly intermediates may also explain and at the same time reconcile salt fragmentation experiments that indicated a stable core for γ-TuRC assembly consisting of GCP4, GCP5, and GCP6, either with[17] or without[6] small amounts of GCP2

and GCP3. The following modular core expansion process (Fig. 6b) is accompanied by stepwise structural reorganization of recruited γ-TuSC units. Interestingly, comparing the conformations of γ-TuSC$_{(1,2)}$ in the wild-type and actin-deficient γ-TuRC, the presence of actin in the γ-TuRC seems to elicit a comparable conformational change in γ-TuSC$_{(1,2)}$ (Fig. 4d) as observed for γ-TuSC$_{(3,4)}$ and γ-TuSC$_{(5,6)}$ during the assembly process (Fig. 3b). This indicates that one of the roles of actin in the complex, although not essential for γ-TuRC assembly, could be to conformationally lock the terminal γ-TuSC$_{(1,2)}$ unit. The most striking structural rearrangement observed during γ-TuRC assembly occurs on the level of the γ-TuSC$_{(13,14)}$ unit in preparation of the final assembly phase. Stable incorporation of the γ-TuSC$_{(13,14)}$ unit, which is likely to repeatedly associate and dissociate over the assembly process, is a two-step process. First, γ-TuSC$_{(13,14)}$ slides into its final position along the GRIP1 domain interface during or right after the formation of the full lumenal bridge. Correlation of these two events during γ-TuRC assembly may indicate a proofreading mechanism for correct lumenal bridge assembly before ring closure, possibly mediated by direct interactions between the lumenal bridge and the GCP3$_{(14)}$ GRIP1 domain, both of which are in close spatial proximity in the locked position of the γ-TuSC$_{(13,14)}$ unit. In the second step, γ-TuSC$_{(13,14)}$ is stabilized by direct binding to the γ-TuSC$_{(1,2)}$ unit, preventing γ-TuSC$_{(13,14)}$ from dissociating and at the same time physically closing the ring (Fig. 6c). Even though recapitulating central steps in γ-TuRC assembly, our structural data do not indicate which steps of the process might be dependent on the AAA+ ATPase RUVBL1/RUVBL2[9]. Dedicated experiments will be required to capture the structure of γ-TuRC assembly intermediates in complex with RUVBL1/RUVBL2 to elucidate how this AAA+ ATPase promotes γ-TuRC assembly.

Already two decades ago, Murphy et al. uncovered that human γ-TuRC contains five paralogous GCP proteins, raising the question of why so many structurally highly similar GCP variants are part of the complex[2]. In our study, we identified GCP variant-specific structural features of GCP4, GCP5, and GCP6 to be centrally involved in the assembly process from both a structural and mechanistic point of view, starting to explain the higher number of GCP paralogs in the vertebrate γ-TuRC, as compared to yeast, where GCP2 and GCP3 are sufficient[21]. The GCP6-specific GRIP1-NTE plays a key role in the overall stabilization of the 6-spoke intermediate by establishing contacts with all γ-TuRC-specific GCP proteins, and in initiating the formation of the lumenal bridge by recruiting the MZT1-GCP3$_{(8)}$ module. Moreover, the GCP6 GRIP1-NTE likely plays a critical role in driving the modular expansion phase and in generating uniform complexes by selectively binding and thereby stabilizing a defined number of γ-TuSC units during the assembly process. The GCP4 and the GCP5 paralogs in combination are likely important for creating and specifically occupying a structurally distinct binding site for MZT1 modules on GCP3$_{(8)}$, which may underlie MZT1-GCP3$_{(8)}$ module availability for lumenal bridge assembly (Fig. 2d).

The identification of binding sites for MZT1-GCP modules on the outer surface of the γ-TuRC, which were not resolved in any previously published cryo-EM reconstruction of native or recombinant γ-TuRC[6–9], was entirely unexpected. While the MZT1-GCP modules may play a role during γ-TuRC assembly by providing extended intersubunit interfaces, we could observe these modules not only in assembly intermediates but also in the fully assembled complex, indicating that their association with the outer surface of the γ-TuRC is not merely an effect of an incomplete assembly process. Instead, alternative patterns of post-translational modifications on the recombinant γ-TuRC could be an important factor in stabilizing or destabilizing the association of the MZT1-modules on the outer surface of the

complex. MZT1-module release could hence serve as a regulatory mechanism with a potential role during γ-TuRC recruitment to the centrosome.

Since the surprising discovery of actin as an integral part of the lumenal bridge[6–9], its function in the context of the γ-TuRC remained a mystery. After specific inhibition of actin integration into the γ-TuRC, we unexpectedly observed that actin was neither required for the assembly of 14-spoked γ-TuRC nor for its structural integrity (Fig. 4b and Supplementary Fig. 12b–f) despite its central position in the complex. This indicates that the assembly defect previously observed when omitting MZT1 and actin from the recombinant expression system[14] was mostly due to the absence of MZT1, not actin, and thus indirectly emphasizes the stabilizing role of the MZT1-GCP modules forming the α-helical scaffold of the lumenal bridge. Analyzing the effect of inhibited actin integration into the γ-TuRC in vivo, we observed that the actin-depleted γ-TuRC is functionally strongly impaired, despite correct assembly on the GCP-γ-tubulin unit level and recruitment to centrosomes. The observed phenotypes might be explained by dysregulated or less effective microtubule nucleation resulting from the altered γ-TuRC geometry observed after inhibiting actin-binding (Fig. 4d), by the involvement of factors in the nucleation reaction that associate with the γ-TuRC in an actin-dependent manner, or potentially by the formation of microtubules with a different protofilament number. Furthermore, actin binding to the γ-TuRC might be required for microtubule nucleation only in certain contexts, for instance during centriole duplication[29]. Thus, the microtubule nucleation reaction in vivo is likely even more complex than anticipated, as underlined by the discrepancy between microtubule nucleation activity of actin-binding-deficient γ-TuRC in vitro and in vivo. The requirement of actin to be stably integrated into the γ-TuRC for full functionality as the principal vertebrate microtubule nucleator, as demonstrated in our study, adds an important facet to the dynamic interplay between actin and microtubule networks at the centrosome, which has started to be recognized as a central process in shaping cell division[30–33].

## Methods

**Cloning for recombinant expression in insect cells.** Cloning of the recombinant γ-tubulin complexes was based on the MultiBac^TM (GENEVA Biotech) expression system and was done as described previously[13]. Briefly, PCR amplified gene of *MZT2B* (Transomic Technologies) and *RUVBL1* and *RUVBL2* gene fragments (IDT, USA) were cloned into MultiBac^TM via InFusion cloning (Takara) using primers listed in Supplementary Table 2. MultiBac^TM plasmids containing the human genes *TUBG1, TUBGCP2, TUBGCP3, TUBGCP4, 2xFLAG-TUBGCP5, TUBGCP6, ACTB,* and *MZT1* were described before[13]. Gene fragments of human *RUVBL1* and *RUVBL2* were optimized for insect cell expression and purchased (IDT, USA) with 5′- and 3′-overhangs (5′-AAAACCTATAAATATG, 3′-TCTA GAGCCTGCAG) for InFusion cloning.

For the construct to express recombinant γ-TuSC, *TUBGCP3* was tagged with a TEV-2xFLAG-Tag on the C-terminus via PCR amplification. Expression cassettes were combined with the same strategy as before using standardized primers Supplementary Table 2. N-terminal deletion (ΔN56-GCP6) was done via PCR amplification with deletion primers (Supplementary Table 2) of *TUBGCP6* in the pIDC plasmid. After PCR, the reaction solution was mixed with *Dpn1* restriction enzyme (NEB), incubated for 1 h at 37 °C and afterwards used for transformation and further cloning steps as for the other constructs. The constructs for baculovirus production were assembled via subsequent Cre-recombination (Cre-recombinase NEB) as described in the MultiBac^TM manual (version 5.1). To generate γ-TuRC^WT and γ-TuRC^ΔN56-GCP6, the following constructs were assembled. Construct 1: pACEBac1 *2xFLAG-TUBGCP5*; pIDC *TUBGCP6*; pIDK *TUBG1, TUBGCP4*; pIDS *ACTB*[13]. construct 2: pACEBac1 *MZT1*; pIDC *TUBGCP2, TUBGCP3*; pIDK *TUBG1, MZT2B*; pIDS *RUVBL1, RUVBL2*. construct ΔN56-GCP6: pACEBac1 *2xFLAG-TUBGCP5*; pIDC *ΔN56-TUBGCP6*; pIDK *TUBG1, TUBGCP4*; pIDS *ACTB*. For the expression omitting GCP2 and GCP3 (γ-TuRC^ΔGCP2/3), construct 3 was produced: pACEBac1 *MZT1, MZT2B*; pIDK *TUBG1*; pIDS *RUVBL1, RUVBL2*. The construct for expression of γ-TuSC consists of: pACEBac1 *MZT1, MZT2B*; pIDC *TUBGCP2, TUBGCP3-2xFLAG*; pIDK *TUBG1*; pIDS *RUVBL1, RUVBL2*. All plasmids were verified via PCR amplification and sequencing.

**Cloning of *E. coli* expression constructs**. The DNA fragments of the N-terminus of human $TUBGCP6_{N126}$ (residues 1–126) and $MZT1$ were amplified via PCR (Q5 DNA Polymerase, NEB, Supplementary Table 2). Using PCR, $TUBGCP6_{N126}$ was tagged with C-terminal 8xHis tag and cloned into the Nde1 and BamH1site in the pET26b expression vector, while $MZT1$ was tagged with an N-terminal FLAG-Tag via PCR and cloned into the Nco1 and EcoR1 cleavage site in the pETDUET-1 dual expression vector. To create a co-expression plasmid, $TUBGCP6_{N126}$-8xHis was amplified via PCR and inserted into the Nde1 and Xho1 cleavage site of pET-DUET1- FLAG-MZT1.

To generate pETDUET-1-FLAG-MZT1-TUBGCP6$_{N126}$ with different point mutations in GCP6$_{N126}$ (R35D, K38D, K39D; R35A, K38A, K39A; Y42A, F46A; R35D, K38D, K39D, Y42A, F46A), mutagenesis was performed as described for pIDC $\Delta$N56-TUBGCP6, utilizing primers listed in Supplementary Table 2.

**Protein expression and purification in *E. coli***. Proteins were expressed in *E. coli* BL21 CodonPlus-RIL (Stratagene) in cultures up to 1 liter using 2xYT medium (Roth). Protein expression was induced with 0.2 mM isopropyl-ß-D-thiogalactopyranoside (IPTG) at an optical density of 0.6–0.8 and performed at 18 °C overnight. Afterwards, expression cultures were divided into several aliquots, harvested by centrifugation, flash frozen, and stored at −80 °C until further use.

**Actin IP experiment**. Cell pellets from 4 ml expression culture were dissolved in 300 µl of IP-lysis buffer (50 mM Tris-HCl pH 8.0, 150 mM NaCl, 1 mM DTT, 1% Triton-X, 5 mM ATP, 5 mM MgCl$_2$, 1:100 PI, 1:100 PMSF) and sonicated (5 × 30 s, Bioruptor). Afterwards, the sample was centrifuged (10 min, 3320 × g, 4 °C), the supernatant was mixed with 20 µl of His Beads (Profinity$^{TM}$ IMAC Ni-Charged Resin, Biorad) and incubated for 1 h at 4 °C. Beads were washed 3x with IP-wash buffer (50 mM Tris-HCl pH 8.0, 150 mM NaCl, 10 mM Imidazole) and equilibrated with 400 µl G-buffer (5 mM Tris-HCl pH 8.0, 0.2 mM CaCl$_2$, 0.2 mM ATP, 0.5 mM DTT). Afterwards, the G-buffer was removed, 100 µl of actin solution (0.1 mg/ml actin protein, from rabbit skeletal muscle, Cytoskeleton, Inc. SKU:AKL99, dissolved in G-buffer) was added and the sample was incubated for 30 min at 4 °C. After the incubation, the sample was washed with 400 µl of ice-cold G-buffer and eluted with 60 µl IP-elution buffer (50 mM Tris-HCl pH 8.0, 150 mM NaCl, 500 mM Imidazole) after 20 min of incubation. Samples from IP experiments, as well as controls, actin only and IPs in absence of beta actin, were mixed with SDS sample buffer and used for immunoblot analysis.

**Recombinant γ-TuRC expression and purification**. Bacmid production was performed as described in the MultiBac$^{TM}$ manual (GENEVA Biotech version 5.1). Bacmids were used for virus production in Sf21 insect cells with cellfectin II reagent (Thermo Fisher Scientific). Viruses were amplified in 30 ml (1 × 10$^6$ cells/ml) and diluted 1:100 in 100 ml (1.5 × 10$^6$ cells/ml) expression culture in Sf21 cells using Sf-900 III medium supplemented with (100 units/ml penicillin/100 µg/ml streptomycin, Thermo Fisher Scientific). Infected cells were kept at 27 °C for 60 h, harvested via centrifugation (800 × g for 5 min), flash frozen in liquid N$_2$ and stored at −80 °C until further usage.

For protein purification, cells were resuspended in 15 ml cold lysis buffer (50 mM Tris, pH 7.5, 200 mM NaCl, 1 mM MgCl$_2$, 1 mM EGTA, 0.5 mM DTT, 0.1% (vol/vol) Tween-20 with 10 µl Benzonase (Sigma Aldrich) and one complete EDTA-free protease inhibitor tablet (ROCHE). Resuspended cells were sonicated (3 × 1 min with 0.6 amplitude, Hielscher UP50H) and centrifuged at 20,000 × g for 30 min at 4 °C. Anti-FLAG M2 Affinity Gel (Sigma-Aldrich) was equilibrated in lysis buffer and incubated with the centrifuged lysate for 90 min with rotation at 4 °C. Beads were separated from lysate by centrifugation (800 × g, 3 min). Afterwards, beads were washed once with lysis buffer (5 ml) and twice (2.5 ml) with wash buffer (50 mM Tris, pH 7.5, 150 mM NaCl, 1 mM MgCl$_2$, 1 mM EGTA, 0.5 mM DTT). After each washing step, beads were sedimented via centrifugation (800 × g, 3 min). Elution was done with one bead volume of elution buffer (wash buffer plus 0.5 mg/ml 3xFLAG peptide (Gentaur)) and 30 min incubation rotating at 4 °C in an Eppendorf tube. Afterwards, elution was separated form beads via centrifugation (800 × g, 3 min). Elution was repeated once, and samples were used for subsequent experiments or flash frozen in liquid N$_2$ and stored at −80 °C. SDS-PAGE were run together with Page Ruler Plus (Thermo Fisher Scientific) on precast gels (10% or 4%–20% BIO-RAD) and stained with Coomassie Brilliant Blue G250 (Sigma Aldrich) or used for immunoblot analysis of the γ-TuRC components.

**Recombinant γ-TuSC expression and purification**. The expression and FLAG purification of recombinant γ-TuSC was done as described for γ-TuRC. FLAG elutions were either used for subsequent experiments or were loaded onto an anion exchange column (Mono Q® 5/50 GL column, Cytiva) equilibrated with wash buffer (50 mM Tris, pH 7.5, 150 mM NaCl, 1 mM MgCl$_2$, 1 mM EGTA, 0.5 mM DTT). Complexes were eluted via a gradient from 150 mM NaCl to 1 M NaCl concentration, over 20 column volumes with a flow rate of 0.4 ml/min. Peak fractions were checked via SDS-PAGE, combined and either used for subsequent experiments or concentrated (Amicon 30 K) to 1.2 mg/ml before flash freezing in liquid N$_2$ and storage at −80 °C. Anion exchange chromatography were run using

Unicorn software (version 7.5, Cytiva) and graphs were plotted using Prism software (GraphPad v. 9.2.0).

**Human stable cell lines generation for in vivo studies**. The Retro-X™ Tet-On® 3G Inducible Expression System (Clontech) was used to generate RPE1-hTERT cells with the Tet-On® 3G System and to integrate the siRNA-resistant *TUBGCP6-C-terminal-FLAG* wild-type and mutant *(A mut (R35A, K38A, K39A); D mut (R35D, K38D, K39E); Y mut (Y42A, F46A); ALL mut (R35D, K38D, K39E, Y42A, F46A); dN (ΔN56-GCP6))* constructs into the cell genome. Silent mutations were introduced into the *TUBGCP6* cDNA sequence (nt 156-171 5′-TGAGACTCAA CAGCTG-3′ was mutated to 5′-cGAaACcCAgCAatTa-3′) to generate siRNA-resistant constructs.

The Retro-X™ Tet-On® 3G Inducible Expression System (Clontech) was also used to generate HEK T293 cells with the Tet-On® 3G System to integrate the *ΔN56-TUBGCP6-FLAG* construct into the genome. Cells were selected in medium containing 0.05 µg/ml of Puromycin (Invivogen Cat# ant-pr-1) for 5 days. Cloning was done as described above using primers listed in Supplementary Table 2. *ΔN-TUBGCP6* RPE1-hTERT KO (referred to as *ΔN-GCP6*) cells were obtained via the CRISPR/Cas9 gene-editing strategy. Two sgRNAs (5′-GGTGTTTTCCCACACGC CGG-3′ and 5′-GCTGGTAGTTTTTGACATGTC-3′) were used to target the Exon 1 in a way to excise the region coding the first 60 amino acids and thus to generate a *GCP6* mutant lacking the actin-binding helices and create a new start codon downstream the excised one (Supplementary Fig. 13g). For this reason, the two sgRNAs were designed to cut shortly upstream the endogenous start codon (−25 nt) and in front of the first methionine codon which could function as a new start codon (+180 nt). The sgRNAs were cloned into the pX458 plasmid[34] and delivered into the cells via electroporation. Single clones were grown separately, then screened via genomic PCR (agarose gels were analyzed with Quantity One v4.6.9, BioRad) and Sanger sequencing using the primer pair 5′-GTGGGCACTT TCCACGGGTGAG-3′ and 5′-GAAGAGCGAGACCCCGGGTG TCT-3′. Sequencing data of the *ΔN-GCP6* knock-out cell lines were analyzed with SnapGene (v5.3.2). *mNeonGreen-LMNB1* and *TUBG1-mRuby2* were integrated into RPE1-hTER wild-type cells and into the *ΔN-GCP6* RPE1-hTERT KO clones using the pQCXIP retroviral system (Clontech).

**siRNA depletion and rescue**. Endogenous GCP6 depletion was accomplished by targeting *TUBGCP6* transcripts with the siRNA 5′-GAUGAGACUCAACAGCU GCUU-3′[6] delivered into the cells via Lipofectamine RNAiMAX Transfection (Thermo Fisher). Cells were analyzed after 60 h of depletion. To rescue the depletion and to analyze the function of the mutants, the *TUBGCP6-C-terminal-FLAG* wild-type and mutant construct expression were induced by adding 1 µg/ml of doxycycline to the cell medium 36 h before the end of the siRNA depletion experiment.

**Microtubule regrowth assay**. Cells were put on ice and let incubate at 4 °C in order to depolymerize microtubules from the centrosome. After 30 min cold treatment, cells were washed with 37 °C 1X PBS and placed on a pre-heated metal rack (at 37 °C) for 5 s (siRNA rescue experiment) or 10 s (KO microtubule-regrowth experiment) to allow microtubule repolymerization. Then, cells were briefly extracted with CSK buffer in order to remove the soluble cytoplasmic non-polymerized components of the cytoskeleton and then the cells were fixed in ice-cold methanol for 5 min at −20 °C prior immunofluorescence (IF). The number of microtubules per centrosome and the microtubule size (length) were quantified using Fiji software[35].

**Immunoblotting, immunofluorescence, and microscopy**. Fixed cells were blocked for 30 min with 10% FBS (v/v), 0.1% Triton-X100 (v/v) in 1X PBS. 1 h incubation with primary antibody was followed by 30 min incubation with secondary AlexaFluor antibody and 4′,6-Diamidine-2′-phenylindole dihydrochloride (DAPI). Antibodies were diluted in 3% Bovine Serum Albumin (BSA, w/v). The samples were mounted on glass slides with Mowiol mounting medium. Antibodies (see Supplementary Table 3) used for protein detection were used in the following dilutions: GAPDH (rabbit, 1:1000, CellSignalling 14C10), Vinculin (mouse, 1:2000, Proteintech 66305-1-Ig), γ-tubulin (ms, 1:1000, Abcam Ab27074), γ-tubulin (guinea-pig, 1:50, homemade[36]), GCP6 (rabbit, 1:1000, Bethyl A302-662A), TUBGCP3 (rabbit, 1:1000, Proteintech 15719-1-AP), GCP4 (rabbit, 1:1000, see Liu et al. 2019), DYKDDDDK (FLAG-tag) (mouse, 1:1600, Cell signaling 9A3), α-tubulin (mouse, 1:500, SigmaAldrich DM1A), BubR1 (mouse, 1:200, Abcam Ab4637), Pericentrin (rabbit, 1:2000, Abcam Ab4448), penta-HIS-HRP (mouse, 1:2000, Proteintech HRP-66005), penta-His (mouse, 1:2000, QIAGEN, 34660), DDDDK tag (FLAG)(rabbit, 1:1000, Proteintech, 20543-1-AP), β-Actin (Mouse, 1:1000, Proteintech 66009-1-lg), α-tubulin (rabbit, 1:1000, MBL PM054). For immunoblot analysis proteins were blotted on PVDF membrane (Merck, cat Nr. IPWH00010) in 25 mM Tris-HCl pH 8.3, 192 mM Glycine buffer and detected with antibodies listed in Supplementary Table 3. Immunoblots were imaged with LAS4000IR (v2.1). Both IF and live-cell imaging data were acquired on a Delta-Vision Elite system (Leica) with an Olympus IX71 microscope equipped with 60X and 100X objective lenses. Live-cell experiment data were acquired by imaging

*wild-type* and *ΔN-GCP6* cells expressing mNeonGreen-LaminB1 and γ-tubulin-mRuby2 with 4 min intervals.

**ΔN56-GCP6-FLAG γ-TuRC purification from HEK T293 cells.** Expression of the *ΔN56-TUBGCP6*-FLAG mutant construct in HEK T293 cells was induced by adding 10 ng/ml doxycycline to the cell medium 48 h before harvesting the cells. Cells were harvested from two 15 cm dishes and the cell pellet was washed with 1X PBS prior lysis. Cell pellets were lysed in lysis buffer (50 mM Na-HEPES pH 8.0, 1 mM EGTA, 1 mM MgCl₂, 100 mM NaCl, 0.1% (vol/vol) TritonX, 0.1 mM GTP and one complete EDTA-free protease inhibitor tablet (ROCHE)) using a Potter-Elvehjem Homogenizer (Thermo scientific) and afterwards centrifuged at 20,000 × *g* for 15 min at 4 °C. The cell lysate was incubated with magnetic FLAG beads (M2, Sigma Aldrich) for 2 h and samples were washed once with lysis buffer and two times with wash buffer (50 mM Na-HEPES pH 8.0, 1 mM EGTA, 1 mM MgCl₂, 100 mM NaCl, 0.1% (vol/vol) TritonX, 0.1 mM GTP) before they were eluted with elution buffer (50 mM Na-HEPES pH 8.0, 1 mM EGTA, 1 mM MgCl₂, 100 mM NaCl, 1 mM GTP, 0.2 mg/ml 3xFLAG peptide (Gentaur)) for 30 min at 4 °C and directly used for negative stain EM analysis.

**In vitro microtubule nucleation assays.** In vitro microtubule nucleation was measured using two different assays. First, a fluorescence-based Tubulin Polymerization assay kit (Cytoskeleton, Denver Com cat. no. BK011P) was used following manufactures instructions (Supplementary Fig. 12g). The concentration of γ-TuRC and γ-TuRC^ΔN56-GCP6 was normalized according to the γ-tubulin signal in immunoblot analysis and the molar concentration was determined by comparing to the signal of a dilution series of recombinant human γ-tubulin[37,38]. γ-TuRC samples were diluted in elution buffer accordingly and measured together with the negative control, elution buffer only, and the positive control, 3 μM paclitaxel. For the measurements, samples were mixed at 4 °C in a 384-well microtiter plate with the reaction buffer (18 μM αβ-tubulin (porcine tubulin), 80 mM PIPES pH 6.9, 2 mM MgCl₂, 0.5 mM EGTA, 1 mM GTP, 15% (w/v) glycerol). Afterwards, the plate was transferred to a prewarmed (37 °C) plate reader chamber to start the reaction. Fluorescence was measured at constant temperature (37 °C) for 60 min in 1 min intervals (CLARIOstar, BMG Labtech, excitation, F: 360–10, emission, F: 450–10).

The second assay (Supplementary Fig. 12h) was based on manual counting of newly nucleated microtubules on light microscopy images as described previously[6]. In brief, the αβ-tubulin concentration was 20 μM, containing 5% Cy3-labelled tubulin for visualization. The molar γ-tubulin concentration of γ-TuRC and γ-TuRC^ΔN56-GCP6 samples was normalized and determined to 7 nM as described above. For each sample (γ-TuRC, γ-TuRC^ΔN56-GCP6, and buffer control), three different time points (0 min, 1 min, and 3 min) after starting the nucleation reaction were analyzed. The number of nucleated microtubules was quantified on 15 images in *n* = 3 replicates for each sample and time point. Data were processed using the Fiji and Prism software (GraphPad v. 9.2.0).

**Negative stain EM.** Five μl of the sample was applied on glow-discharged copper-palladium 400-mesh EM grids covered with an approximately 10 nm thick continuous carbon layer. After 30 s of incubation at room temperature, grids were blotted with a Whatman filter paper 50 (CAT N.1450-070) and washed on three drops of water. The sample on grids was stained with 3% uranyl acetate in water. Negative stain EM data for 2D class averaging were acquired on a Talos L120C TEM equipped with 4 k × 4 K Ceta CMOS camera (Thermo Fisher Scientific). Data were acquired using EPU (Thermo Fischer Scientific) at a nominal defocus of approximately −2 μm and an object pixel size of 0.2556 nm (Supplementary Fig. 8f, g), 0.328 nm (Supplementary Fig. 6d, e; Supplementary Fig. 9; Supplementary Fig. 12e, f; Supplementary Fig. 13e) or 0.4125 nm (Supplementary Fig. 1d, e). Micrographs shown in Supplementary Figures were adjusted in brightness and contrast using Fiji software.

**Negative stain EM data processing.** Image processing for all datasets was performed in Relion 3.1[39]. The contrast transfer function (CTF) of micrographs was estimated using Gctf[40]. For all datasets, either approximately 500 particles were selected manually to create an initial 2D class for automated particle picking, or an already generated 2D class of the same data type was used. Particles were extracted, either scaled (5.11 Å pixel size: γ-TuSC, Supplementary Fig. 8; 6.56 Å pixel size: γ-TuSC oligomerization, γ-TuRC^ΔGCP2/3, γ-TuRC^ΔN56-GCP6, Supplementary Figs. 6, 9 and 12; 8.25 Å pixel size: wild-type γ-TuRC, Supplementary Fig. 1) or at full spatial resolution (2.556 Å pixel size: γ-TuSC, Supplementary Fig. 8; 3.28 Å pixel size: γ-TuSC oligomerization, γ-TuRC^ΔGCP2/3, γ-TuRC^ΔN56-GCP6, Supplementary Figs. 6, 9, 12, and 13; 4.125 Å pixel size: sample wild-type γ-TuRC, Supplementary Fig. 1), and subjected to 2D classification into 20–200 classes, using a T-factor of 2, a translational search range of 20 pixels at 2 pixels increment and mask with 400 Å diameter.

For wild-type γ-TuRC (Supplementary Fig. 1), 1,197,806 particles were picked on 876 images and subjected to five successive rounds of 2D classification. For γ-TuRC^ΔN56-GCP6 from the insect cell expression system (Supplementary Fig. 12), 194,874 particles were picked on 267 images and subjected to four successive rounds of 2D classification. For γ-TuSC (Supplementary Fig. 8), 268,126 particles

were picked on 530 images and subjected to two successive rounds of 2D classification. For the γ-TuRC^ΔGCP2/3 (Supplementary Fig. 6), 183,636 particles were picked on 329 images and subjected to three successive rounds of 2D classification. The 4-spoke assemblies and single spokes were selected and classified separately. For the γ-TuSC dilution series (Supplementary Fig. 9), 100 images were acquired for each dilution and processed as follows. 1:2 dilution: 134,246 particles sorted in one round of 2D classification; 1:5 dilution: 122,712 particles sorted in two initial rounds of 2D classification, followed by two subsequent rounds of 2D classification with the selected γ-TuSC classes and several rounds of 2D classification with the selected oligomer classes; 1:10 dilution: 110,642 particles sorted in two initial rounds of 2D classification, followed by two subsequent rounds of 2D classification with the selected γ-TuSC classes and several rounds of 2D classification with the selected oligomer classes; 1:20 dilution: 76,325 particles sorted in two initial rounds of 2D classification, followed by two subsequent rounds of 2D classification with the selected γ-TuSC classes and several rounds of 2D classification with the selected oligomer classes; 1:50 dilution: 65,732 particles sorted in three successive rounds of 2D classification; 1:100 dilution: 51,950 particles sorted in six successive rounds of 2D classification. For γ-TuRC^ΔN56-GCP6 from HEK T293 cells, 110,331 particles were picked on 319 micrographs and subjected to four successive rounds of 2D classification.

**Cryo-EM sample preparation and data acquisition.** Four μl of purified recombinant human wild-type γ-TuRC, recombinant human γ-TuRC^ΔN56-GCP6 or recombinant human γ-TuRC^ΔGCP2/3 was applied on Quantifoil holey carbon grids (Cu; R2/1; 200 mesh) that were glow-discharged beforehand in a Gatan Solarus 950 (Gatan, Inc.) plasma cleaner for 30 s. Grids were blotted for 0.5 s immediately after sample application and subsequently plunge frozen into liquid ethane using a Vitrobot Mark IV (Thermo Fisher Scientific). Cryo-EM data were acquired on a Titan Krios transmission electron microscope (Thermo Fisher Scientific) operated at 300 kV and equipped with a Quanta GIF energy filter and K3 (Gatan, Inc.) direct detector operated in dose fractionation mode (50 frames/frame stack for wild-type γ-TuRC and γ-TuRC^ΔN56-GCP6, 40 frames/frame stack for recombinant human γ-TuRC^ΔGCP2/3). Data were acquired at a pixel size of 2.66 Å (wild-type γ-TuRC and γ-TuRC^ΔN56-GCP6) or 1.07 Å (γ-TuRC^ΔGCP2/3) using the EPU (Thermo Fischer Scientific) 'fast acquisition scheme'. For recombinant human γ-TuRC, the cumulative dose was ~35 e⁻/Å² at a dose rate of 25 e⁻/px/s. For recombinant human γ-TuRC^ΔN56-GCP6, the cumulative dose was ~42.6 e⁻/Å² at a dose rate of 30 e⁻/px/s. For recombinant human γ-TuRC^ΔGCP2/3, the cumulative dose was ~62.4 e⁻/Å² at a dose rate of 25 e⁻/px/s. Data for wild-type γ-TuRC and γ-TuRC^ΔN56-GCP6 were collected at a nominal defocus range from −2 to −2.5 μm with 1 frame stack in the centre of each hole. Data for recombinant human γ-TuRC^ΔGCP2/3 were collected at a nominal defocus range from −1 to −3 μm with four frame stacks per hole.

**Cryo-EM data processing for recombinant wild-type γ-TuRC.** The initial processing steps were performed in Relion 3.1[39]. For 6,127 frame stacks, beam-induced motion was corrected in MotionCor2[41] and the CTF was estimated in Gctf[40]. For particle picking, Topaz[42] was trained on 1,241 picked particles from 50 micrographs with the resnet8 convolutional neural network model using scaling factor 3 and expecting 100 particles in average on each micrograph. Topaz picked 436,887 particles that were extracted with a box size of 200 pixels and downscaled to 128 pixels (scaled pixel size: 3.9219 Å). Particles were split into four random subsets that were subjected to 3D classification into six classes, using a T-factor of T = 20, a spherical mask of 400 Å and a translational search of 20 pixels with 2-pixel step. In total, 10 classes containing 159,770 particles were selected from all 4 subsets. Particles were recentered and re-extracted at full spatial resolution in boxes of 200 pixels and subsequently subjected to a second round of 3D classification using parameters as described above, except for translational search that was limited to 5 pixels with 1-pixel step. Next, particles corresponding to larger (10–14 spokes; 36,881 particles) and smaller (6–10 spokes; 92,149 particles) assemblies were selected separately and further processed in Relion 3.0 Beta[39]. Both sets of particles were separately subjected to 3D autorefinement, subsequent CTF refinement, and Bayesian polishing[43]. This was followed by a second round of 3D refinement and subsequent multibody refinement, using three to five bodies depending on the number of spokes in the respective assembly intermediate. The individual segments obtained from multibody refinement[44] were merged in UCSF Chimera[45] and subjected to post-processing in Relion, reaching global resolution of 7.5 Å for the large spoke assemblies and 5.3 Å for the small spoke assemblies (Supplementary Fig. 3a, b).

Fragmented density for the peripheral γ-TuSC units in these two initial reconstructions indicated substoichiometry or conformational heterogeneity. In the next steps, we therefore focused on comprehensive sub-classification of particle sets to reach a complete coverage of structurally homogenous γ-TuRC assembly states. All subsequent classification steps were performed with a T-factor of T = 20, a spherical mask of 400 Å, 3–6 classes and without translational and orientational sampling. First, the set of particles representing 'small' assemblies was computationally sorted according to the presence of γ-TuSC₍₅,₆₎ and γ-TuSC₍₁₃,₁₄₎ in two separate classification rounds. Different combinations of particles originating from these two classification runs (+/− γ-TuSC₍₅,₆₎ and +/− γ-TuSC₍₁₃,₁₄₎) were subjected to 3D autorefinement and multibody refinement and resulted in the following reconstructions of homogenous assembly intermediates: spokes 7–12 (7 Å resolution; Supplementary Fig. 3d), spokes

7–14 (7.6 Å resolution; Supplementary Fig. 3e), spokes 5–12 (8.6 Å resolution; Supplementary Fig. 3f) and spokes 5–14 (9.0 Å resolution; Supplementary Fig. 3g). Second, the set of particles representing 'large' assemblies was computationally sorted according to the presence of γ-TuSC$_{(1,2)}$ and γ-TuSC$_{(13,14)}$ in two separate classification rounds. Different combinations of particles originating from these two classification runs (+/− γ-TuSC$_{(1,2)}$ and +/− γ-TuSC$_{(13,14)}$) were subjected to 3D autorefinement and multibody refinement and resulted in the following reconstructions of homogeneous assembly intermediates: spokes 3–12 (8.1 Å resolution; Supplementary Fig. 3h), spokes 3–14 (9.0 Å resolution; Supplementary Fig. 3i), spokes 1–12 (9.2 Å resolution; Supplementary Fig. 3j) and spokes 1–14 (16.1 Å resolution without multibody refinement; Supplementary Fig. 3k).

Furthermore, to obtain reconstructions of 'large' assembly intermediates with the focus set on γ-TuSC$_{(1,2)}$ at the highest possible level of detail, we selected particles including or lacking γ-TuSC$_{(1,2)}$—independent of the presence or absence of γ-TuSC$_{(13,14)}$—and subjected them to 3D autorefinement and multibody refinement, obtaining the following reconstructions: 'large' assembly including γ-TuSC$_{(1,2)}$ (8.7 Å; Supplementary Fig. 3l) and 'large' assembly lacking γ-TuSC$_{(1,2)}$ (7.8 Å; Supplementary Fig. 3m). Similarly, we separately refined a particle set originating from the classification focused on γ-TuSC$_{(13,14)}$—independent of the presence or absence of γ-TuSC$_{(1,2)}$—to obtain a reconstruction of the 'large' assembly, including γ-TuSC$_{(13,14)}$ in the 'locked' conformation (8.7 Å; Supplementary Fig. 3n).

**Cryo-EM data processing for recombinant γ-TuRC$^{ΔN56-GCP6}$.** Pre-processing of 2,614 frames stacks, automatic particle picking, particle extraction, and initial 3D classification of 448,184 particles in two subsets was performed as described for recombinant wild-type γ-TuRC. Assembly intermediates were omitted and in total, two classes containing 41,054 particles representing the fully assembled γ-TuRC$^{ΔN56-GCP6}$ were selected from the two subsets. The retained particles were recentered, re-extracted at the full spatial resolution, and subjected to a second round of 3D classification using parameters as described above to remove the remaining false-positive particles. five classes containing 40,115 particles were selected and further processed in Relion 3.0 Beta, as described for recombinant wild-type γ-TuRC to obtain a reconstruction of recombinant γ-TuRC$^{ΔN56-GCP6}$ at 7.1 Å global resolution (Supplementary Fig. 3o). All FSC curves were plotted in Gnuplot v5.2[46].

**Cryo-EM data processing for recombinant γ-TuRC$^{ΔGCP2/3}$.** Pre-processing of 11,276 frames stack was performed as described for recombinant wild-type γ-TuRC. After CTF estimation, micrographs were split into two subsets for auto-mated particle picking by Topaz using the same training parameters as for recombinant wild-type γ-TuRC. The number of picked particles from the two subsets was 936,634 and 951,532 particles, respectively.

The following processing steps were performed in CryoSPARC[47]. Both particle sets were separately subjected to 2D classification with 200 classes and a circular mask diameter of 250 Å. 24,357 particles contained in the best two 2D class averages representing the 4-spoke assembly intermediate were selected and subjected to ab-initio reconstruction. The resulting density was used as a reference for homogeneous refinement, which resulted in a reconstruction with three clearly resolved spokes and additional density segments corresponding to a fourth more flexible spoke.

This reconstruction was used as a reference for subsequent 3D classification in Relion 3.1 with parameters as described for recombinant wild-type γ-TuRC using an extended set of 53,311 particles originating from the initial 2D classification in cryoSPARC (see above). Using the resulting best 3D class as a reference, all ~1.9 million particles located by Topaz (split into several subsets) were subjected to four consecutive rounds of 3D classification with a shape mask and T-factor of 20. We retained a final set of 9,192 particles that were further processed in Relion 3.0 Beta as for the recombinant wild-type γ-TuRC (see above), and obtained final reconstructions at 8.5 Å and 7.8 Å global resolution before and after multibody refinement (body 1: GCP4-5-4, body 2: GCP6), respectively.

**Model building and refinement.** Atomic models for all cryo-EM reconstructions were composed from previously published models of γ-TuRCs. GCP2, GCP3, GCP4, γ-tubulins, and actin were obtained from the atomic model of the recombinant human γ-TuRC (PBD 7AS4)[9]. GCP5 and GCP6 were taken from the atomic model of human native γ-TuRC (PDB 6V6S)[8]. The GCP5 (210–266) and GCP6 GRIP1 NTEs (282–351) were built in Coot[48] according to previously published cryo-EM densities (EMD-21074, EMD-21069)[8]. Components of the lumenal bridge and the MZT1-GCP5 module were taken from previously published atomic models of the native human γ-TuRC (PDB 6X0U)[10], the X-ray structure of the human MZT1-GCP5 module (PDB 6L81)[11] and the X-ray structure of the human MZT1-GCP6 module (PDB 6M33)[11], respectively.

These components were fitted as rigid bodies into all cryo-EM reconstructions separately using UCSF Chimera[45]. GRIP1 and GRIP2 domains of GCP proteins were fitted as separate rigid bodies where possible. Several C-termini of GCP2 (873–876), the GCP5 N-terminal segment (106–113) and the GCP3 N-terminal segment (119–130) were extended in Coot according to densities with alanines.

Rigid body-fitted models were subjected to molecular dynamics flexible fitting (MDFF) performed in VMD v1.9.3[49] except the model of 4-spoke assembly intermediate. The plugin QwikMD[50] was used for the preparation of the model.

The plugin MDFF v0.4[51] was used to generate the namd configuration file. The MDFF run was submitted with NAMD v2.14[52] using 2000 minimization steps, 20000 simulation steps in vacuum and applying a grid force of 0.3. Flexibly fitted models were subjected to Phenix[53] real-space refinement running in 2 macrocycles. MDFF of the rigid body-fitted model of 4-spoke assembly intermediate was performed in Namdinator v2.0[54].

Vector representations of conformational changes were prepared in PyMOL (PyMOL v2.1, Schrödinger). Visual representations of cryo-EM densities were prepared in Chimera or ChimeraX[55].

**Statistics.** All relevant information on the statistical analysis of the experiments is provided in the figure legends. Data in graphs are represented as mean±s.d. of at least 3 replicates. $p$ values of experiments were derived from two-tail unpaired $t$-test analysis with a significance level of $p ≤ 0.05$. Statistical analysis was done using Prism (GraphPad v. 9.2.0) and Microsoft Excel (v16.46.21021202).

**Reporting summary.** Further information on research design is available in the Nature Research Reporting Summary linked to this article.

## Data availability

The data that support this study are available from the corresponding authors upon reasonable request. Atomic coordinates and the associated cryo-EM densities for assembly intermediates of the recombinant wild-type γ-TuRC have been deposited in the Protein Data Bank and the Electron Microscopy Data Bank under accession codes 7QJ0/EMD-14005, 7QJ1/EMD-14006, 7QJ2/EMD-14007, 7QJ3/EMD-14008, 7QJ4/EMD-14009, 7QJ5/EMD-14010, 7QJ6/EMD-14011, 7QJ7/EMD-14012, 7QJ8/EMD-14013, 7QJ9/EMD-14014, 7QJA/EMD-14015, 7QJB/EMD-14016, 7QJC/EMD-14017. Atomic coordinates and the associated cryo-EM densities for recombinant actin-binding-deficient γ-TuRC and the recombinant wild-type 4-spoke assembly intermediate have been deposited in the Protein Data Bank and the Electron Microscopy Data Bank under accession codes, PDB-7QJD/EMD-14018, PDB-7QJE/EMD-14019, respectively. The raw cryo-EM micrograph movie stacks are available from the corresponding authors only upon request. Previously reported structural data used in this article: 6X0U, 6L81, 6M33, 6V6S, 7AS4, EMD-21074, EMD-21069, EMD-11888. Source data are provided with this paper.

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

## Acknowledgements

The authors thank Ursula Jäkle (ZMBH, Heidelberg), Teresa Debatin (IPMB, Heidelberg), Olga Kolesnikova (EMBL, Heidelberg) for support during molecular cloning, protein expression and purification, Lukas Rohland for support during microtubule nucleation measurements and Till Rudack (Ruhr University, Bochum) for support with atomic modeling. We thank the light microscopy Imaging Facility (Holger Lorenz, ZMBH, Heidelberg) and the Protein Expression and Purification Core Facility (EMBL, Heidelberg) for support. We acknowledge the services SDS@hd and bwHPC supported by the Ministry of Science, Research and the Arts Baden-Württemberg, as well as the German Research Foundation (INST 35/1314-1 FUGG and INST 35/1134-1 FUGG). We also acknowledge access to the infrastructure of the Cryo-EM Network at the Heidelberg University (HDcryoNET) and support by Dirk Flemming (BZH, Heidelberg) and Götz Hofhaus (BioQuant, Heidelberg). This work is supported by grants of the Deutsche Forschungsgemeinschaft (DFG) to E.S. (DFG Schi 295/4-4) and to S.P. (DFG PF 963/1-4).

## Author contributions

M.W. and E.Z. constructed and purified samples for cryo-EM, performed cryo-EM data collection and together with G.T. and B.J.A.V. analyzed the cryo-EM data. M.W. performed cloning, protein expression, and protein purification. E.Z. acquired cryo-EM data, analyzed cryo-EM densities, and created models and figure representations. E.S.A. created constructs, generated human cell lines and performed in vivo experiments and light microscopy data acquisition and analyzed together with A.B. and M.W. light microscopy data. A.N. performed negative staining and data acquisition. A.N., M.W., B.J.A.V., E.Z. analyzed negative stain E.M. data. A.B. performed microtubule nucleation assay. A.S.R., A.B., performed actin IP experiments. S.E., E.S. and S.P. supervised the experiments. M.W., E.Z., E.S.A., E.S. and S.P. wrote the manuscript. All authors discussed the data and gave final approval for publication.

## Funding

## Competing interests

The authors declare no competing interests.
