## [Peer Review File · Nature Communications]

Modular assembly of the principal microtubule nucleator γ -TuRCReviewers' Comments:

Reviewer #1:

Remarks to the Author:

Würtz et al present a very elegant, comprehensive study of the stepwise assembly of γ -TuRC. The work substantially advances our mechanistic understanding of this elaborate process, by demonstrating in ample detail the likely sequence of molecular events leading to γ -TuRC formation, including key conformational changes and inter-molecular dependencies enabling the correct, functional assembly of γ -TuRC.

The authors explain specific roles of various γ -TuRC components and neatly show their function-serving dynamicity. They find the GCP2-3-4-5-4-6 6-spoke assembly to be a cornerstone for the whole 14-spoke structure and that the N-terminal extension of the GRIP1 domain of the GCP6 paralog is critical for the stabilisation of the 6-spoke intermediate. They map the topology of the previously unresolved MZT1-containing modules and provide a new insight into the so far poorly understood, or perhaps even misunderstood, function of the single actin molecule that is bound within the γ -TuRC lumen.

In summary, through their extensive work Würtz et al identify key players and interactions governing the assembly of the fundamental "molecular platform" for microtubule nucleation. I think Nature Communications is a suitable journal for this work and I enthusiastically support its publication therein.

The authors might consider further comments below useful for the revision.

Minor points:

1. I am wondering about microtubule (MT) protofilament (pf) architecture nucleated from the actin-less γ TuRC. If it's different than physiological, the in-vitro MT polymerisation assay will not pick it up, but the in-vivo functional assay shows a disfunction, which may be related to a switch in a dominating pf number. Maybe a point for discussion and/or the future cryo-EM studies? (This would not require 3D; 2D moire-pattern analyses in moderately highly defocused raw images should suffice to quantify this)

2. I think a few figure callouts are muddled up - you may want to double-check throughout the manuscript. I noted potential errors in the following:

My best guesses (or "?" when not sure) in the following chunks of text - please, check if useful:

-Top of p. 4: "Unexpectedly, we identified three previously unresolved MZT1-GCP3 modules (Fig. 1a,c; Supplementary Fig. 4a) and one MZT1-GCP5 module (see below) on the outer surface of the γ -TuRC, associated with the GRIP2 domains of GCP3 subunits (Fig. 2c? or 1a, right)."

-Middle of p.9: "We obtained a cryo-EM reconstruction of fully assembled γ -TuRC Δ N56-GCP6 (Fig. 4b) at 7.1 Å global resolution (Supplementary Fig. 10b)"

-Middle of p.14: "Interestingly, comparing the conformations of γ -TuSC(1,2) in the wild-type and actin-deficient γ -TuRC (Fig.? and 4d)"

3. I note that near-atomic resolution was in principle not necessary for the main focus of the study, but I am still slightly pondering the pixel size that appears large for today's standards (2.66 Å). It may be limiting at least for the highest-resolution structure obtained (~5.3 Å). Any particular reason for this pixel size?

4. Not sure what the authors meant in the first sentence of the second paragraph of the Results section (p.3), starting with: "To ensure correct GCP paralog incorporation..." - please, consider revising.
5. Would recommend rounding resolution claims to the first decimal place (e.g. bottom of p.3: "at 7.49 Å global resolution..." - would change to 7.5 Å).
6. Bottom of p.4: "allowed us to unambiguously identify the unassigned density" - feels like too strong a statement based on data depiction (Supplementary Fig. 4c) - likely, but not unambiguous - I'd suggest softening.
7. Middle of p.6: "In particular, the GRIP2 domains and the associated γ -tubulin copies are displaced from their canonical position" - perhaps a figure callout might be useful there.
8. I'd suggest the following small edits in the middle of p.10:

"Having observed actin-dependent changes in γ -TuRC geometry, we next tested the microtubule nucleation efficiency of γ -TuRC Δ N56-GCP6 in vitro using a batch fluorescence-based nucleation assay. In this assay, γ -TuRC Δ N56-GCP6 nucleated microtubules to a similar extent as the recombinant wild-type γ -TuRC (Supplementary Fig. 11g), indicating that actin is not essential for microtubule formation under the nucleation conditions of the in vitro experiment.

We next assessed what effect the actin-binding-deficient γ -TuRC has on microtubule nucleation and cell viability under physiological conditions in vivo" - this reads better as a next paragraph to me.

(Please, ignore bits of text that should be sub/superscripted - change unintended)
9. Top of p.15: "Since the surprising discovery of actin as an integral part of the luminal bridge" - would suggest adding a reference here.
10. Would suggest the following small edit at the bottom of p.15: "The requirement of actin to be stably integrated into the γ -TuRC for full functionality as the principal vertebrate microtubule nucleator, as demonstrated in our study..."
11. Would suggest dropping the brackets for figures/tables citations in such cases: "...using primers listed in Supplementary Table 1" - example from the first paragraph of the Methods section, top of p.24.
12. An (annoyingly) obvious question: any particular reason to switch from Relion 3.1 to 3.0 beta during the processing?
13. I appreciate all FSC curves are scale-matched in the Supplementary Fig. 3, but I think it is more important to show the curve crossing the 0.143 threshold and coming down to 0 in panel b.
14. Supplementary Fig. 4b: the label above the structures is probably wrong (copied from panel a?)
15. Lastly, I'd recommend checking definite and indefinite articles throughout the manuscript.

Yours faithfully,
Szymon W. Manka

Reviewer #2:

Remarks to the Author:

This is a detailed study of the assembly pathway of gamma-tubulin ring complexes (gamma-TuRCs) from recombinant components. Structural intermediates of gamma-TuRC assembly are identified and characterized by cryo-EM. The data indicate a 6-spoke intermediate with a GCP4/5/4/6 core that has one gamma-TuSC attached. It is demonstrated that this intermediate associates with additional gamma-TuSCs to form full-sized gamma-TuRCs. This part of the manuscript is an extension of earlier findings by the same authors, and by Haren et al (2020, J Cell Sci), who had shown in biochemical experiments the existence of a similar assembly mechanism from a GCP4/5/4/6 core by deconstructing native gamma-TuRCs. The important novel information in this manuscript is on the order of gamma-TuSC integration into the gamma-TuRC, with particular roles of the gamma-TuSCs in positions 13,14 and 1,2. Moreover, this study identifies the positions of previously unresolved Mzt1 modules, associated with GCPs 3 and 5. Another very interesting contribution is the demonstration that recombinant gamma-TuRCs can assemble in the absence of actin, using an N-terminally truncated form of GCP6 that is deficient in actin binding, and that cells with such incomplete gamma-TuRCs show spindle defects in mitosis.

Overall, I think that this is a very well controlled, complete study that extends significantly our knowledge on the assembly and the structure of gamma-TuRCs. This manuscript should be of high interest to a wide readership working on microtubule nucleation. I have no particular criticism.

Reviewer #3:

Remarks to the Author:

Würtz and co authors investigated structural factors underlying the assembly of the human γ -tubulin ring complex (γ TuRC). The authors use a combination of cryo-EM reconstruction and computational processing to provide additional details to the assembly pathway of the γ TuRC and the recently discovered contribution of actin in the luminal bridge. The authors purified human γ TuRCs using the eukaryotic insect cell expression system and affinity purification and obtained cryo-EM reconstructions of the sample. The dataset was subject to computational particle sorting and 3D classification to uncover the structure of multiple intermediates of the fully assembled γ TuRC. The authors further probe the cryo-EM densities of these intermediates to investigate structural characteristics of each to recapitulate a possible assembly process. The process they describe begins with an intermediate consisting of 6-spokes which includes one γ TuSC unit combined with ring-complex specific GCP proteins. The authors report the identification of three previously unresolved MZT1-GCP3 modules and one MZT1-GCP5 module. As well, they observe 'conformational' locking of subunits including a geometric shift of γ TuSC and completion of the luminal bridge upon recruitment of γ TuSC; and the locking of the repeatedly associating and dissociating γ TuSCs. Finally, the authors attempt to determine the function of the actin component of the γ TuRC luminal bridge, and rather surprisingly find that while actin is not required for γ TuRC assembly or microtubule nucleation in vitro, cells expressing a GPC6 mutation that depletes actin from the luminal bridge exhibit defects in microtubule regrowth and errors in mitosis that are consistent with reduced activity of the γ TuRC in vivo.

- Overall, the authors provide a logical framework for the proposed modular assembly pathway. This information is very interesting, but the study does not make a convincing case that the intermediates found in their dataset and the interactions that they detail represent real biological phenomena that occur in vivo and the significance for the mechanism. Biological relevance for the proposed mechanism is lacking in the discussion, which instead focuses on structure-function relationships for each step.

- The authors provide evidence that the nucleation activity of the γ TuRC when depleted of actin is reduced, in contrast to the similar level of in vitro microtubule nucleation of the mutation that blocks actin binding. One concern is the lack of details for the batch-nucleation assays in both the results

section and methods, which do not provide the concentrations of γ TuRC (in nM?) and $\alpha\beta$ tubulin (in μ M?) in the reaction along with other important information, and the rationale for including paclitaxel is not explained.

- The authors should measure the nucleation activity of the γ TuRC and γ TuRC Δ N56 GCP6 using single molecule measurements as recently reported by Wieczorek et al. (JCB 2021) to accurately determine relative activity, as even minor changes in microtubule number could have significant effects in vivo.

- Does depletion of actin in vivo and inhibition of actin binding in vitro result in the same outcome in a population of γ TuRCs. This is an important question that the authors should consider as the interpretation of the in vivo results is complex.

Point-by-Point Reply to Reviewers

We thank the reviewers for their thoughtful and very supportive comments. Below we address the specific points raised by the reviewers and elaborate on the corresponding changes in the manuscript.

Importantly, we have added a series of new *in vitro* microtubule nucleation experiments, as well as new cryo-EM and negative stain EM analyses in response to the comments of Reviewer #3.

Reviewer #1 (Remarks to the Author):

Würtz et al present a very elegant, comprehensive study of the stepwise assembly of γ -TuRC. The work substantially advances our mechanistic understanding of this elaborate process, by demonstrating in ample detail the likely sequence of molecular events leading to γ -TuRC formation, including key conformational changes and inter-molecular dependencies enabling the correct, functional assembly of γ -TuRC.

The authors explain specific roles of various γ -TuRC components and neatly show their function-serving dynamicity. They find the GCP2-3-4-5-4-6 6-spoke assembly to be a cornerstone for the whole 14-spoke structure and that the N-terminal extension of the GRIP1 domain of the GCP6 paralog is critical for the stabilisation of the 6-spoke intermediate. They map the topology of the previously unresolved MZT1-containing modules and provide a new insight into the so far poorly understood, or perhaps even misunderstood, function of the single actin molecule that is bound within the γ -TuRC lumen.

In summary, through their extensive work Würtz et al identify key players and interactions governing the assembly of the fundamental "molecular platform" for microtubule nucleation. I think Nature Communications is a suitable journal for this work and I enthusiastically support its publication therein.

We thank the Reviewer for the very positive evaluation of our manuscript.

The authors might consider further comments below useful for the revision.

Minor points:

1. I am wondering about microtubule (MT) protofilament (pf) architecture nucleated from the actin-less γ TuRC. If it's different than physiological, the *in-vitro* MT polymerisation assay will not pick it up, but the in-vivo functional assay shows a disfunction, which may be related to a switch in a dominating pf number. Maybe a point for discussion and/or the future cryo-EM studies? (This would not require 3D; 2D moire-pattern analyses in moderately highly defocused raw images should suffice to quantify this)

We thank the Reviewer for this insightful comment. We agree that the number of protofilaments could potentially be relevant for the MT polymerization defects we observe *in vivo* for actin binding-deficient γ -TuRC, because MT-binding proteins that are involved in MT nucleation and elongation most likely have a preference for MTs comprising of exactly 13 protofilaments.

As the Reviewer points out, we think that pursuing structural analysis of microtubule minus ends nucleated by the actin binding-deficient γ -TuRC *in vitro* and *in vivo*, including the relevant control experiments, is going beyond the scope of this study.

As suggested by the Reviewer, we have included a comment addressing this aspect in the discussion section on p. 16:

“The observed phenotypes might be explained by dysregulated or less effective microtubule nucleation resulting from the altered γ -TuRC geometry observed after blocking actin binding (Fig. 4d), by involvement of factors in the nucleation reaction that associate with the γ -TuRC in an actin-dependent manner, or potentially by formation of microtubules with a different protofilament number.”

2. I think a few figure callouts are muddled up - you may want to double-check throughout the manuscript. I noted potential errors in the following:

My best guesses (or "?" when not sure) in the following chunks of text - please, check if useful:

We have double-checked that figure callouts are correct and adjusted the incorrect callouts as suggested by the Reviewer.

-Top of p. 4: "Unexpectedly, we identified three previously unresolved MZT1-GCP3 modules (Fig. 1a,c; Supplementary Fig. 4a) and one MZT1-GCP5 module (see below) on the outer surface of the γ -TuRC, associated with the GRIP2 domains of GCP3 subunits (Fig. 2c? or 1a, right)."

Adjusted to: "...associated with the GRIP2 domains of GCP3 subunits (1a, right)."

-Middle of p.9: "We obtained a cryo-EM reconstruction of fully assembled γ -TuRC Δ N56-GCP6 (Fig. 4b) at 7.1 Å global resolution (Supplementary Fig. 10b)"

Adjusted to: "We obtained a cryo-EM reconstruction of fully assembled γ -TuRC Δ N56-GCP6 (Fig. 4b) at 7.1 Å global resolution (Supplementary Figs. 3o, 10b)."

-Middle of p.14: "Interestingly, comparing the conformations of γ -TuSC(1,2) in the wild-type and actin-deficient γ -TuRC (Fig.? and 4d)"

We removed this figure callout entirely, because the relevant figure panels (Fig. 3b and 4d) are called in the second half of the sentence.

3. I note that near-atomic resolution was in principle not necessary for the main focus of the study, but I am still slightly pondering the pixel size that appears large for today's standards (2.66 Å). It may be limiting at least for the highest-resolution structure obtained (~5.3 Å). Any particular reason for this pixel size?

We fully agree with the Reviewer that the pixel size we chose was limiting the resolution of the most abundant assembly intermediate, the 6-spoke assembly. However, we found the pixel size chosen to be a good tradeoff between the level of detail required to address the open questions on the one hand and the comparably low overall particle density, as well as the high compositional heterogeneity of the sample on the other hand.

4. Not sure what the authors meant in the first sentence of the second paragraph of the Results section (p.3), starting with: "To ensure correct GCP paralog incorporation..." - please, consider revising.

Given the complexity of the recombinant expression system, our first aim was to confirm that all five different GCP paralogs assemble in the correct order and stoichiometry. We have revised the sentence as follows:

“To confirm correct assembly of the γ -TuRC in our recombinant expression system...”

5. Would recommend rounding resolution claims to the first decimal place (e.g. bottom of p.3: "at 7.49 Å global resolution..." - would change to 7.5 Å).

We have corrected the manuscript accordingly.

6. Bottom of p.4: "allowed us to unambiguously identify the unassigned density" - feels like too strong a statement based on data depiction (Supplementary Fig. 4c) - likely, but not unambiguous - I'd suggest softening.

We note that the identification of the MZT1-GCP5 module in the cryo-EM density of the 6-spoke assembly was indeed unambiguous based on the rigid body fits shown in Supplementary Fig. 4c. We have now chosen better viewing angles and density threshold levels in the figure to support the assignment, but additionally have softened the text as suggested:

“... allowed us to identify the unassigned density...”

7. Middle of p.6: "In particular, the GRIP2 domains and the associated γ -tubulin copies are displaced from their canonical position" - perhaps a figure callout might be useful there.

We thank the Reviewer for this suggestion and have included a callout to Fig. 3b, in which the displacement is shown.

8. I'd suggest the following small edits in the middle of p.10:

"Having observed actin-dependent changes in γ -TuRC geometry, we next tested the microtubule nucleation efficiency of γ -TuRC Δ N56-GCP6 in vitro using a batch fluorescence-based nucleation assay. In this assay, γ -TuRC Δ N56-GCP6 nucleated microtubules to a similar extent as the recombinant wild-type γ -TuRC (Supplementary Fig. 11g), indicating that actin is

not essential for microtubule formation under the nucleation conditions of the in vitro experiment.

We next assessed what effect the actin-binding-deficient γ -TuRC has on microtubule nucleation and cell viability under physiological conditions in vivo" - this reads better as a next paragraph to me.

(Please, ignore bits of text that should be sub/superscripted - change unintended)

We thank the Reviewer for the suggestions and modified the text accordingly.

9. Top of p.15: "Since the surprising discovery of actin as an integral part of the luminal bridge" - would suggest adding a reference here.

We thank the Reviewer for this suggestion and have included references to all four publications originally identifying actin as part of the γ -TuRC.

10. Would suggest the following small edit at the bottom of p.15: "The requirement of actin to be stably integrated into the γ -TuRC for full functionality as the principal vertebrate microtubule nucleator, as demonstrated in our study..."

We have revised the text according to the Reviewer's suggestion.

11. Would suggest dropping the brackets for figures/tables citations in such cases: "...using primers listed in Supplementary Table 1" - example from the first paragraph of the Methods section, top of p.24.

We have removed the brackets in all similar instances through the entire manuscript.

12. An (annoyingly) obvious question: any particular reason to switch from Relion 3.1 to 3.0 beta during the processing?

Switching from Relion 3.1 to Relion 3.0 has a very practical background: The default reading mode for MRC micrograph movie stacks in Relion switched from 'unsigned' to 'signed' format with the version update from 3.0 to 3.1, apparently to increase compatibility to IMOD. As a result, the Relion 3.1 particle polishing algorithm could not handle our micrograph movie stacks in MRC 'unsigned' format anymore and we switched back to Relion 3.0 to successfully complete particle polishing.

13. I appreciate all FSC curves are scale-matched in the Supplementary Fig. 3, but I think it is more important to show the curve crossing the 0.143 threshold and coming down to 0 in panel b.

We are actually showing the full FSC plots in all panels of Supplementary Fig. 3. The FSC curve in panel b does not cross the threshold criterion and does not drop to 0, because the cryo-EM density is resolved to Nyquist frequency in this case due to the high number of contributing particles and the comparably large pixel size (see comment 3).

14. Supplementary Fig. 4b: the label above the structures is probably wrong (copied from panel a?)

We have double-checked the labeling in Supplementary Fig. 4b. We are showing the rigid body fit of either the MZT1-GCP3 module (panel a) or the MZT2-GCP2 module (panel b) into the same density segment associated with GCP3₍₆₎ (both panels), so the labeling is correct.

15. Lastly, I'd recommend checking definite and indefinite articles throughout the manuscript.

We thank the Reviewer for this suggestion. We have double-checked and revised several occurrences of definite and indefinite articles in the text.

Reviewer #2 (Remarks to the Author):

This is a detailed study of the assembly pathway of gamma-tubulin ring complexes (gamma-TuRCs) from recombinant components. Structural intermediates of gamma-TuRC assembly are identified and characterized by cryo-EM. The data indicate a 6-spoke intermediate with a GCP4/5/4/6 core that has one gamma-TuSC attached. It is demonstrated that this intermediate associates with additional gamma-TuSCs to form full-sized gamma-TuRCs. This part of the manuscript is an extension of earlier findings by the same authors, and by Haren et al (2020, J Cell Sci), who had shown in biochemical experiments the existence of a similar assembly mechanism from a GCP4/5/4/6 core by deconstructing native gamma-TuRCs. The important novel information in this manuscript is on the order of gamma-TuSC integration into the gamma-TuRC, with particular roles of the gamma-TuSCs in positions 13,14 and 1,2. Moreover, this study identifies the positions of previously unresolved Mzt1 modules, associated with GCPs 3.

Another very interesting contribution is the demonstration that recombinant gamma-TuRCs can assemble in the absence of actin, using an N-terminally truncated form of GCP6 that is deficient in actin binding, and that cells with such incomplete gamma-TuRCs show spindle defects in mitosis.

Overall, I think that this is a very well controlled, complete study that extends significantly our knowledge on the assembly and the structure of gamma-TuRCs. This manuscript should be of high interest to a wide readership working on microtubule nucleation. I have no particular criticism.

We thank the Reviewer for the very positive evaluation of our manuscript and we are pleased that there are no specific comments to be addressed.

Reviewer #3 (Remarks to the Author):

Würtz and co authors investigated structural factors underlying the assembly of the human γ -tubulin ring complex (γ TuRC). The authors use a combination of cryo-EM reconstruction and computational processing to provide additional details to the assembly pathway of the γ TuRC and the recently discovered contribution of actin in the luminal bridge. The authors purified human γ TuRCs using the eukaryotic insect cell expression system and affinity purification and obtained cryo-EM reconstructions of the sample. The dataset was subject to

computational particle sorting and 3D classification to uncover the structure of multiple intermediates of the fully assembled γ TuRC. The authors further probe the cryo-EM densities of these intermediates to investigate structural characteristics of each to recapitulate a possible assembly process. The process they describe begins with an intermediate consisting of 6-spokes which includes one γ TuSC unit combined with ring-complex specific GCP proteins.

The authors report the identification of three previously unresolved MZT1-GCP3 modules and one MZT1-GCP5 module. As well, they observe ‘conformational’ locking of subunits including a geometric shift of γ TuSC and completion of the luminal bridge upon recruitment of γ TuSC; and the locking of the repeatedly associating and dissociating γ TuSCs. Finally, the authors attempt to determine the function of the actin component of the γ TuRC luminal bridge, and rather surprisingly find that while actin is not required for γ TuRC assembly or microtubule nucleation *in vitro*, cells expressing a GPC6 mutation that depletes actin from the luminal bridge exhibit defects in microtubule regrowth and errors in mitosis that are consistent with reduced activity of the γ TuRC *in vivo*.

- Overall, the authors provide a logical framework for the proposed modular assembly pathway. This information is very interesting, but the study does not make a convincing case that the intermediates found in their dataset and the interactions that they detail represent real biological phenomena that occur *in vivo* and the significance for the mechanism. Biological relevance for the proposed mechanism is lacking in the discussion, which instead focuses on structure-function relationships for each step.

The presence of γ -TuRC subcomplexes has been solidly established using biochemical approaches in the past. Endogenous γ -TuRC subcomplexes can be identified on a regular basis in sucrose gradient analysis experiments of cell extracts (Choi et al., 2010; Haren et al., 2020; Murphy et al., 2001). Furthermore, salt fragmentation experiments were performed by others and our lab, indicating that assembly of the γ -TuRC starts from a stable core containing GCP4, GCP5 and GCP6 (Haren et al., 2020; Liu et al., 2019).

All assembly intermediates we identify during cryo-EM analysis correspond to correctly assembled subcomplexes of the γ -TuRC and no subcomplexes with ‘unnatural’ GCP variant order can be observed. This clearly indicates that the assembly intermediates characterized in our study represent *bona fide* steps in cellular biogenesis of the γ -TuRC.

In the revised manuscript, we have further substantiated this with a new cryo-EM experiment, in which we structurally analyze the 4-spoke assembly intermediate obtained when GCP2 and GCP3 were not co-expressed. Based on structural features, we find that the 4-spoke intermediate indeed corresponds to a GCP4-5-4-6 assembly, thus identifying another γ -TuRC subcomplex directly reflecting the molecular architecture of the fully assembled complex in our recombinant expression system and further emphasizing the contribution of GCP variant-specific properties for the assembly process. This new experiment is now summarized in the main text and shown in Supplementary Fig. 7.

Cumulatively, these data clearly indicate that the assembly mechanism characterized in our study is biologically relevant and recapitulated *in vivo*.

As suggested by the Reviewer, we now also elaborate on these points in the discussion section of the manuscript on p. 14: “Importantly, all of these assembly intermediates represent subcomplexes of the γ -TuRC directly reflecting the molecular architecture of the fully assembled complex. This clearly indicates that the assembly intermediates characterized in

this study represent *bona fide* steps in cellular biogenesis of the γ -TuRC, as frequently observed using biochemical approaches, for instance sucrose gradient analysis (Choi et al., 2010; Haren et al., 2020; Murphy et al., 2001) or salt fragmentation experiments (Haren et al., 2020; Liu et al., 2019) .”

- The authors provide evidence that the nucleation activity of the γ TuRC when depleted of actin is reduced, in contrast to the similar level of *in vitro* microtubule nucleation of the mutation that blocks actin binding. One concern is the lack of details for the batch-nucleation assays in both the results section and methods, which do not provide the concentrations of γ TuRC (in nM?) and $\alpha\beta$ tubulin (in μ M?) in the reaction along with other important information, and the rationale for including paclitaxel is not explained.

In response to this comment, we now provide a more detailed description of the experiments. In particular, the $\alpha\beta$ -tubulin concentration is now given in μ M in the methods section and the γ -tubulin concentrations, as determined by immunoblotting using a dilution series of recombinant γ -tubulin as reference, are given in the figure legend of Supplementary Fig. 12g. Furthermore, we have explained the rationale for including paclitaxel, which serves as a positive control, as part of the kit for the microtubule nucleation reaction, in the methods section.

We further extended the set of *in vitro* microtubule nucleation experiments using different concentrations of γ -TuRC to dissect in more detail the kinetics of microtubule nucleation by the γ -TuRC ^{Δ N56-GCP6} mutant compared to wild-type γ -TuRC. For these experiments, all relevant information is included in the figure legend of Supplementary Fig. 12 and the methods section.

- The authors should measure the nucleation activity of the γ TuRC and γ TuRC Δ N56 GCP6 using single molecule measurements as recently reported by Wieczorek et al. (JCB 2021) to accurately determine relative activity, as even minor changes in microtubule number could have significant effects *in vivo*.

In response to this Reviewer comment, we performed a series of additional experiments to more accurately dissect the nucleation behavior of wild-type γ -TuRC and γ -TuRC ^{Δ N56-GCP6} summarized below:

1) We extended the set of *in vitro* batch-nucleation experiments to dissect in more detail the kinetics of microtubule nucleation by using a range of γ -TuRC and TuRC ^{Δ N56-GCP6} concentrations now covering a broad spectrum of microtubule nucleation activities. However, under none of the conditions we could observe a measurable difference in microtubule nucleation activity. These data are included in Supplementary Fig. 12g of the revised manuscript.

2) We performed an alternative *in vitro* nucleation experiment, which we included in Supplementary Fig. 12h of the revised manuscript. By pelleting and counting *in vitro* nucleated microtubules as described before (Liu et al., 2019) instead of measuring batch fluorescence, we determined the number of nucleated microtubules more directly for different time points after starting the nucleation reaction. Even using this approach of determining the microtubule nucleation activity, we could not detect any measurable difference between wild-type γ -TuRC and γ -TuRC ^{Δ N56-GCP6}.

Importantly, even using TIRF single molecule measurements as suggested by the Reviewer, Wieczorek et al. (JCB 2021) do not observe any differences in the *in vitro* microtubule nucleation activity of recombinant wild-type γ -TuRC and the γ -tubulin ‘half rings’ they obtained after omitting MZT1 and actin from the expression system. Since these ‘half rings’ are structurally much more impaired than our Δ N56-GCP6 mutant γ -TuRC, it is highly unlikely that differences in microtubule nucleation activity *in vitro* would become evident for our Δ N56-GCP6 mutant γ -TuRC from single molecule TIRF measurements as reported in Wieczorek et al. (JCB 2021).

In our study, we therefore decided to focus on characterizing the effects of blocking actin binding to the γ -TuRC *in vivo*, where we indeed can observe a robust effect. Negative stain-EM and 2D class averaging of γ -TuRC ^{Δ N56-GCP6} purified from HEK T293 cells (performed in response to comment 4 of Reviewer #3) further validate these *in vivo* data by demonstrating that microtubules are nucleated by structurally intact mutant γ -TuRC in our human cell line model.

The defects in microtubule nucleation of γ -TuRC ^{Δ N56-GCP6} we observe *in vivo* could either be related to a difference in activity too small to be measured by *in vitro* nucleation experiments, as suggested by the Reviewer, or it could be due to a more complex regulation process *in vivo* that is not being recapitulated *in vitro*.

We address all of the aspects discussed above in a new paragraph of the results sections: “Having observed changes in γ -TuRC geometry dependent on integration of actin, we next tested the nucleation efficiency of γ -TuRC ^{Δ N56-GCP6} using two complementary *in vitro* microtubule nucleation assays. In the first assay, nucleation-dependent batch fluorescence was measured over a range of γ -TuRC concentrations (Supplementary Fig. 12g). In the second assays, nucleated microtubules were pelleted and counted on light microscopy images to achieve a more direct readout at early time points after starting the nucleation reaction (Supplementary Fig. 12h). In both assays, γ -TuRC ^{Δ N56-GCP6} nucleated microtubules to a similar extent as the recombinant wild-type γ -TuRC and thus, under the nucleation conditions of the *in vitro* experiments, an impact of γ -TuRC-associated actin on microtubule formation could not be observed. This is in direct agreement with single molecule TIRF experiments (Wieczorek et al., 2021), in which a similar microtubule nucleation efficiency was observed for wild-type γ -TuRC and structurally highly impaired MZT1- and actin-deficient γ -tubulin complexes, recapitulating only half of the γ -tubulin ring. This outcome indicates that the microtubule nucleation process *in vitro* is rather indiscriminative with respect to the structure of the γ -tubulin complex template, but changes in nucleation activity *in vivo* cannot be excluded due to the involvement of additional co-factors such as XMAP215 (Consolati et al., 2020; Thawani et al., 2018, 2020).”

- Does depletion of actin *in vivo* and inhibition of actin binding *in vitro* result in the same outcome in a population of γ TuRCs. This is an important question that the authors should consider as the interpretation of the *in vivo* results is complex.

This comment is ambiguous, because it is not clear whether the reviewer suggests to deplete actin from cells or asks for characterization of the actin binding-deficient γ -TuRC we used for our *in vivo* analysis in human cells. This ambiguity likely goes back to some inaccuracy of how we referred to removal of actin from the γ -TuRC in the manuscript, which in some

instances we term ‘actin depletion from γ -TuRC’. In these instances, actin depletion does not imply that we depleted actin from the cells completely, but that we specifically inhibit binding of actin to the ring complex by either deleting or mutating the two N-terminal α -helices of the GCP6 subunit. In the manuscript, we have removed this ambiguity by referring to ‘actin binding-deficiency of the γ -TuRC’ or ‘inhibited actin incorporation into the γ -TuRC’, instead of ‘actin depletion’.

Thus, there are two alternatives of how to interpret this comment, which we have addressed independently:

Alternative 1) The reviewer suggests to test whether γ -TuRC ^{Δ N56-GCP6} is correctly assembled in the human cell line model, similar to what is observed for the insect cell expression system. In both systems, the very same or highly similar N-terminal deletions of GCP6 (Δ 56 and Δ 60), as well as selected point mutations predicted to minimally perturb the GCP6 fold, were used for γ -TuRC expression, so differences in γ -TuRC structure and molecular architecture are highly unlikely, which is supported by correct recruitment of Δ N56-GCP6 to centrosomes and the very similar levels of γ -tubulin at centrosomes in *GCP6* and Δ N56-*GCP6* cells (Supplementary Fig. 13c,h-j).

However, to also experimentally confirm correct assembly of γ -TuRC ^{Δ N56-GCP6} in the human cell line model in response to this Reviewer comment, we isolated actin binding-deficient γ -TuRC from HEK T293 cells via FLAG purification of Δ N56-GCP6-FLAG and characterized it structurally using negative stain electron microscopy and 2D class averaging. As expected, the resulting 2D classes are indistinguishable from the 2D classes obtained for recombinant γ -TuRC ^{Δ N56-GCP6} from the insect cell expression system, which demonstrates that mutant γ -TuRC is correctly assembled in human cells and strengthens our *in vivo* analysis of microtubule nucleation activity.

These new data have been included in Supplementary Fig. 13e and are being discussed in the respective section of the main text: “Moreover, mutant γ -TuRC ^{Δ N56-GCP6} purified from HEK T293 cells via Δ N56-*GCP6*-FLAG pulldown (see methods) was analysed via negative stain-EM and 2D class averaging (Supplementary Fig. 13e), demonstrating that γ -TuRC ^{Δ N56-GCP6} are intact in the human cell line model and their overall structure is indistinguishable from γ -TuRC ^{Δ N56-GCP6} purified from the recombinant insect cell expression system (Supplementary Fig. 12e,f). Thus, actin binding-deficient GCP6 mutant γ -TuRCs are structurally intact in human cells, but show reduced microtubule nucleation activity *in vivo*.”

Alternative 2) The reviewer suggests to completely deplete actin from cells and subsequently characterize γ -TuRCs from these cells. To address this alternative, we have tried to deplete β - and γ -actin, which have both been observed to be integrated into the γ -TuRC, by siRNA-mediated gene silencing. However, in HEK T293 cells, silencing efficiency for actin was too low to draw any definite conclusions (Figure 1 of the rebuttal letter, see below). Furthermore, actin depletion is known to trigger a variety of secondary effects related to secretion, dynein accumulation, transcription, nuclear organization and protein synthesis that highly complicate interpretation of any observable differences between actin-depleted and control cells.

siRNA beta actin: β -CYA duplex 2, AAACCUAACUUGCGCAGAA

siRNA gamma actin: γ -CYA duplex 2, GAGCCGUGUUUCCUCCAU

NSC: Non specific control

Figure 1. Actin depletion from HEK T293 cells by siRNA-mediated gene silencing. Actin depletion was performed with the indicated siRNA's for 3 days or 4.5 days, and actin levels were compared to a non-specific siRNA control (NSC).

References:

- Choi, Y.-K., Liu, P., Sze, S.K., Dai, C., and Qi, R.Z. (2010). CDK5RAP2 stimulates microtubule nucleation by the gamma-tubulin ring complex. *J. Cell Biol.* *191*, 1089–1095.
- Consolati, T., Locke, J., Roostalu, J., Chen, Z.A., Gannon, J., Asthana, J., Lim, W.M., Martino, F., Cvetkovic, M.A., Rappsilber, J., et al. (2020). Microtubule Nucleation Properties of Single Human γ -TuRCs Explained by Their Cryo-EM Structure. *Dev. Cell.*
- Haren, L., Farache, D., Emorine, L., and Merdes, A. (2020). A stable core of GCPs 4, 5 and 6 promotes the assembly of γ -tubulin ring complexes. *J. Cell Sci.*
- Liu, P., Zupa, E., Neuner, A., Böhler, A., Loerke, J., Flemming, D., Ruppert, T., Rudack, T., Peter, C., Spahn, C., et al. (2019). Insights into the assembly and activation of the microtubule nucleator γ -TuRC. *Nature.*
- Murphy, S.M., Preble, A.M., Patel, U.K., O'Connell, K.L., Dias, D.P., Moritz, M., Agard, D., Stults, J.T., and Stearns, T. (2001). GCP5 and GCP6: two new members of the human gamma-tubulin complex. *Mol. Biol. Cell* *12*, 3340–3352.
- Thawani, A., Kadzik, R.S., and Petry, S. (2018). XMAP215 is a microtubule nucleation factor that functions synergistically with the gamma-tubulin ring complex. *Nat. Cell Biol.* *20*, 575–585.
- Thawani, A., Rale, M.J., Coudray, N., Bhabha, G., Stone, H.A., Shaevitz, J.W., and Petry, S. (2020). The transition state and regulation of γ -TuRC-mediated microtubule nucleation revealed by single molecule microscopy. *Elife* *9*, e54253.
- Wieczorek, M., Ti, S.-C., Urnavicius, L., Molloy, K.R., Aher, A., Chait, B.T., and Kapoor, T.M. (2021). Biochemical reconstitutions reveal principles of human γ -TuRC assembly and function. *J. Cell Biol.* *220*.

Reviewers' Comments:

Reviewer #1:

Remarks to the Author:

The authors responded to all my comments and questions satisfactorily and I am impressed with how they addressed requests from the other reviewers. I recommend publication.

Reviewer #3:

Remarks to the Author:

The authors have addressed my concerns.